# REDUCING CLASS-WISE PERFORMANCE DISPARITY VIA MARGIN REGULARIZATION

**Beier Zhu**[1], **Kesen Zhao**[2], **Jiequan Cui**[3], **Qianru Sun**[4], **Yuan Zhou**[2], **Xun Yang**[1], **Hanwang Zhang**[2]
[1]University of Science and Technology of China [2]Nanyang Technological University,
[3]Hefei University of Technology [4]Singapore Management University
`beier.zhu@ustc.edu.cn`

## ABSTRACT

Deep neural networks often exhibit substantial disparities in class-wise accuracy, even when trained on class-balanced data—posing concerns for reliable deployment. While prior efforts have explored empirical remedies, a theoretical understanding of such performance disparities in classification remains limited. In this work, we present `Margin Regularization` for performance disparity `Reduction` ($MR^2$), a theoretically principled regularization for classification by dynamically adjusting margins in both the logit and representation spaces. Our analysis establishes a margin-based, class-sensitive generalization bound that reveals how per-class feature variability contributes to error, motivating the use of larger margins for "hard" classes. Guided by this insight, $MR^2$ optimizes per-class logit margins proportional to feature spread and penalizes excessive representation margins to enhance intra-class compactness. Experiments on seven datasets—including ImageNet—and diverse pre-trained backbones (MAE, MoCov2, CLIP) demonstrate that our $MR^2$ not only improves overall accuracy but also significantly boosts "hard" class performance without trading off "easy" classes, thus reducing the performance disparity. Codes are available in `https://github.com/BeierZhu/MR2`.

## 1 INTRODUCTION

Deep neural networks have achieved strong generalization across a wide range of domains and tasks. However, as noted by Cui et al. (2024), substantial disparity in class-wise accuracy persist—even when training on class-balanced data. As illustrated in Figure 1(a), models such as ResNet-50 (He et al., 2016a), and ViT-B/16 (Dosovitskiy et al., 2021) trained on ImageNet (Deng et al., 2009) exhibit extreme class-wise accuracy imbalance: for instance, with ResNet-50, the top-performing class reaches 100% top-1 accuracy, while the worst performs at only 16%. The under-performance of certain categories threatens the safe deployment of deep models, making it imperative to design frameworks that combine high accuracy with balanced class-wise performance.

Several works have investigated the causes and remedies for class-wise performance disparity. For instance, Balestriero et al. (2022) shows that common techniques such as data augmentation and weight decay are unfair across classes and can lead to severe performance degradation on certain classes. Cui et al. (2024) attributes the disparity to the problematic representation rather than classifier bias—unlike in long-tailed classification (Menon et al., 2021; Ren et al., 2020; Kang et al., 2020)—and advocates appropriate data augmentation (Yun et al., 2019; Cubuk et al., 2019) and representation learning (*e.g.*, He et al. (2020; 2022)) to mitigate class-wise unfairness. However, these approaches remain largely empirical and lack theoretical grounding. In this work, we provide a principled explanation and framework for reducing class-wise performance disparity.

Our theoretical analysis is both motivated by and consistent with the empirical finding by Cui et al. (2024), which reveals that "hard" (under-performing) classes exhibit greater feature variability than "easy" (well-performing) ones. This is reflected in Figure 1(b), where the $L_2$ norm of the per-class feature variance, increases from "easy" to "hard" classes. In this work, we derive a novel margin-based, class-sensitive learning guarantee (Proposition 1) that connects per-class feature variability and margin design. The result reveals that classes with larger feature variability contribute more to the generalization error. Guided by our theoretical results, we propose `Margin Regularization` for

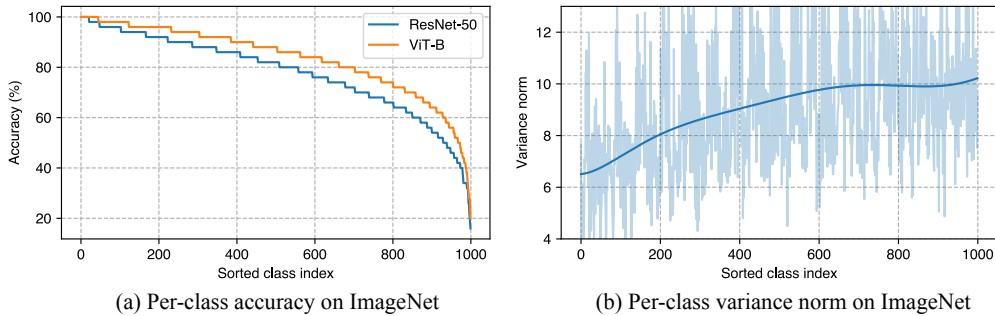

(a) Per-class accuracy on ImageNet          (b) Per-class variance norm on ImageNet

Figure 1: (a) Deep models exhibit severe class-wise accuracy disparity on class-balanced dataset such as ImageNet. (b) Feature diversity imbalance: feature distribution of "hard" classes is more diverse than that of "easy" classes. Classes are sorted in descending order of per-class accuracy.

performance disparity Reduction, dubbed as $\texttt{MR}^2$, a framework that dynamically adjusts margins in both logit and representation spaces during training (Section 3). The logit margin is designed proportional to the feature spread to reduce the generalization bound (Corollary 1), while the representation margin regularization further tightens this bound (Section 4.2). Intuitively, the logit margin loss naturally promotes larger margins for "hard" classes, which are known to improve generalization (Bartlett et al., 1998; Koltchinskii & Panchenko, 2002), and the representation margin loss encourages intra-class feature compactness.

To validate the effectiveness of our $\texttt{MR}^2$, we conduct extensive experiments on seven datasets including ImageNet (Deng et al., 2009) using both convolutional networks (He et al., 2016a;b) and Vision Transformer (Dosovitskiy et al., 2021) architectures. In addition, we evaluate $\texttt{MR}^2$ on diverse pre-trained backbones with both end-to-end fine-tuning and linear probing paradigms—including MAE (He et al., 2022), MoCov2 (He et al., 2020), and CLIP (Radford et al., 2021)—to assess its broad applicability on recent foundation models. We make the following observations: (1) **reduced performance disparity**: our $\texttt{MR}^2$ achieves clear improvements on "hard" classes across all models and datasets, effectively narrowing the performance gap with "easy" classes; and (2) **improved overall-performance**: our $\texttt{MR}^2$ attains moderate gains on "easy" classes as well, indicating that $\texttt{MR}^2$ reduces disparity without sacrificing "easy" class performance. This differs markedly from reweighting methods in debiasing (Nam et al., 2020; Liu et al., 2021), margin-based methods (Cao et al., 2019; Ren et al., 2020), distributionally robust optimization (Sagawa et al., 2020; Jung et al., 2023), etc., which entail performance trade-offs across "hard" and "easy" groups (Section 5.1).

## 2   RELATED WORK

**Margin-based long-tail learning.** In long-tailed classification, where the training data exhibit a skewed label distribution, introducing asymmetric margins into the loss function has proven to be an effective strategy. Typical methods include LDAM (Cao et al., 2019), EQL (Tan et al., 2020), Balanced Softmax (Ren et al., 2020), Logit Adjustment (Menon et al., 2021), GLA (Zhu et al., 2023), PPA (Zhu et al., 2025) and DRO-LT (Samuel & Chechik, 2021). However, these losses are specifically designed for class-imbalanced settings and reduce to the standard cross-entropy loss when class priors are uniform. In contrast, our margin-based loss provides meaningful regularization for improving balanced per-class performance under balanced class priors. Hyperspherical learning methods (Zhou et al., 2022; Son et al., 2025) regulate class or sample margins, often adapting them to data difficulty and class frequency. However, these approaches do not address class-wise performance disparities, which is the central focus of our work.  A detailed discussion is deferred to Section 3.1.

**Mitigating performance disparity in classification.** Recently, several studies (Cui et al., 2024; Balestriero et al., 2022; Li & Liu, 2023; Ma et al., 2022; Jin et al., 2025) have explored ways to address the imbalance accuracies across classes on balanced data. Such work primarily focuses on fairness under adversarial attacks (Li & Liu, 2023; Ma et al., 2022; Jin et al., 2025) or conducts purely empirical analyses of image recognition task (Cui et al., 2024; Balestriero et al., 2022). In contrast, we provide a general class-sensitive analysis that links class-level unfairness to imbalanced

intra-class diversity. Building on this, we derive principled margin regularization at both the logit and feature levels to promote balanced class-wise generalization.

## 3 METHOD

**Setup.** Considering a classification problem with inputs $\mathbf{x} \in \mathcal{X}$ and labels $y \in \mathcal{Y} = [K]$ drawn from an unknown distribution $\mathbb{P}$ with balanced class priors, *i.e.*, $\mathbb{P}(y) = 1/K$. Let $\mathcal{F}$ be a hypothesis set of functions mapping from $\mathcal{X} \times \mathcal{Y}$ to the prediction score (logit) $\mathbb{R}$. For a hypothesis $f \in \mathcal{F}$, the predicted label $\mathsf{f}(\mathbf{x})$ of $\mathbf{x}$ is assigned to the one with the highest score, *i.e.*, $\mathsf{f}(\mathbf{x}) = \mathrm{argmax}_{y \in \mathcal{Y}} f(\mathbf{x}, y)$. Let $\mathbb{1}(\cdot)$ be the indicator function, we define the risk of $f$ as the 0-1 loss over $\mathbb{P}$: $\mathcal{R}(f) = \mathbb{E}_{\mathbf{x}, y \sim \mathbb{P}}[\mathbb{1}(y \neq \mathsf{f}(\mathbf{x}))]$ and the best-in-class risk as $\mathcal{R}^*(\mathcal{F}) = \inf_{f \in \mathcal{F}} \mathcal{R}(f)$. Our goal is to learn a function $f^*$ that attains the best-in-class risk $\mathcal{R}^*$. However, as the distribution $\mathbb{P}$ is unknown, the risk of a hypothesis is not directly accessible. The learner instead measures the empirical risk given a training dataset $\mathcal{D} = \{(\mathbf{x}_i, y_i)\}_{i=1}^N \sim \mathbb{P}^N$: $\widehat{\mathcal{R}}_\mathcal{D}(f) = \frac{1}{N} \sum_{\mathbf{x}, y \in \mathcal{D}} [\mathbb{1}(y \neq \mathsf{f}(\mathbf{x}))]$.

**Notations.** Assume that the function $f$ consists of a feature encoder $\phi : \mathcal{X} \to \mathbb{R}^d$ and a classification head parameterized by $K$ weight vectors $\{\mathbf{w}_k\}_{k=1}^K$. For input $\mathbf{x}$, the feature representation is $\phi(\mathbf{x})$ and the logit for class $y$ is represented as: $f(\mathbf{x}, y) = \mathbf{w}_y^\top \phi(\mathbf{x})$. Let $\mathcal{D}_y$ denote the subset that contains all instances $\mathbf{x}$ belonging to the class label $y$ and $N_y = N/K$ the sample size of class $y$. The empirical per-class mean is computed as $\hat{\boldsymbol{\mu}}_k = \frac{1}{N_y} \sum_{\mathbf{x} \in \mathcal{D}_y} [\phi(\mathbf{x})]$ and the mean squared deviation is computed as $\|\hat{\mathbf{s}}_k\|_2^2 = \frac{1}{N_y} \sum_{\mathbf{x} \in \mathcal{D}_y} [\|\phi(\mathbf{x}) - \hat{\boldsymbol{\mu}}_k\|_2^2]$. During training, we maintain exponential moving average (EMA) of the squared norm of mean $\{\|\hat{\boldsymbol{\mu}}_k\|_2^2\}_{k=1}^K$ and mean squared deviation $\{\|\hat{\mathbf{s}}_k\|_2^2\}_{k=1}^K$. We now present $\mathtt{MR}^2$, a framework that adjusts margins in both logit and representation space.

**Logit margin.** Let $\mathbf{1}_y$ denote the one-hot vector for class $y$, and $\mathbf{z} = [f(\mathbf{x}, 1), ..., f(\mathbf{x}, K)]^\top$ corresponding logit vector. The $\boldsymbol{\gamma}$-margin cross-entropy loss $\ell_{\boldsymbol{\gamma}, \mathsf{ce}} : \mathcal{F} \times \mathcal{X} \times \mathcal{Y} \to \mathbb{R}^+$ for input $\mathbf{x}$ with label $y$ is:

$$\ell_{\boldsymbol{\gamma}, \mathsf{ce}}(f, \mathbf{x}, y) = -\mathbf{1}_y^\top \ln[\mathrm{softmax}(\mathbf{z}/\gamma_y)], \text{ where } \gamma_y = \frac{\bar{c} \cdot K(\|\hat{\boldsymbol{\mu}}_y\|_2^2 + \|\hat{\mathbf{s}}_y\|_2^2)^{\frac{1}{3}}}{\sum_{k=1}^K (\|\hat{\boldsymbol{\mu}}_k\|_2^2 + \|\hat{\mathbf{s}}_k\|_2^2)^{\frac{1}{3}}}, \tag{1}$$

where $\bar{c} > 0$ is a hyper-parameter and $\boldsymbol{\gamma} = [\gamma_1, ..., \gamma_K]^\top$ specifies per-class margins, normalized such that the mean $\bar{\gamma} = \frac{1}{K} \sum_k \gamma_k = \bar{c}$. Note that when $\gamma_k = 1$ for all $k \in [K]$, the $\boldsymbol{\gamma}$-margin cross-entropy loss reduces to the canonical multi-class cross-entropy loss.

At a high level, hard classes (*i.e.*, those with low accuracy) typically exhibit greater feature variability, as observed in both our analysis (Figure 1(b)) and Cui et al. (2024). Intuitively, our loss adaptively assigns larger margins to such classes, leveraging the well-established fact that larger margins lead to better generalization (Bartlett et al., 1998). Therefore the proposed loss *help improve the performance on hard classes.* The per-class margin $\gamma_k$ in Eq. (1) is formally justified in Corollary 1, where we show that it *also minimizes the generalization risk bound.* (The proof is deferred for clarity of exposition.)

**Representation margin.** Let $\bar{s} = \frac{1}{K} \sum_k \|\hat{\mathbf{s}}_k\|_2^2$ denote the average of the mean squared deviation. The representation margin loss $\ell_{\bar{s}} : \mathcal{F} \times \mathcal{X} \times \mathcal{Y} \to \mathbb{R}^+$ is:

$$\ell_{\bar{s}}(f, \mathbf{x}, y) = \ln \left[ 1 + \sum_{\mathbf{x}^+ \in \mathcal{D}_y \backslash \{\mathbf{x}\}} \exp(\|\phi(\mathbf{x}) - \phi(\mathbf{x}^+)\|_2^2 - 2\bar{s}) \right]. \tag{2}$$

Intuitively, $\ell_{\bar{s}}$ encourages intra-class compactness with the parameter $2\bar{s}$ serving as the margin to control the slack. The loss $\ell_{\bar{s}}$ can be regarded as a differentiable relaxation of $\hat{\ell}_{\bar{s}}$ (proof in Section B.1):

$$\hat{\ell}_{\bar{s}}(f, \mathbf{x}, y) = \max \left[ 0, \max_{\mathbf{x}^+ \in \mathcal{D}_y \backslash \{\mathbf{x}\}} \left( \|\phi(\mathbf{x}) - \phi(\mathbf{x}^+)\|_2^2 - 2\bar{s} \right) \right], \tag{3}$$

which encourages the pair-wise distance within the same class no more than $2\bar{s}$. Observe that reducing the pairwise distance is equivalent to minimizing the mean squared deviation, as $\mathbb{E}[\|\phi(\mathbf{x}^+) - \phi(\mathbf{x})\|_2^2] = 2\|\mathbf{s}_k\|_2^2$ (proof in Section B.2). This also explains the origin of the factor 2 that appears in front of $\bar{s}$. In Section 4.2, our theoretical results detail that penalizing large mean squared deviation

helps tighten the generalization error bound. The overall objective of our $\text{MR}^2$ is:

$$\min_{f \in \mathcal{F}} \frac{1}{N} \sum_{\mathbf{x},y \in \mathcal{D}} [\ell_{\boldsymbol{\gamma},\text{ce}}(f,\mathbf{x},y) + \lambda \cdot \ell_{\bar{s}}(f,\mathbf{x},y)], \tag{4}$$

where $\lambda > 0$ is a hyper-parameter that controls the strength of representation regularization.

### 3.1 DISCUSSION WITH EXISTING WORK

This section compares our two losses ($\ell_{\boldsymbol{\gamma},\text{ce}}$ and $\ell_{\bar{s}}$) with prior methods that modify margins at both the logit and representation levels, respectively.

**Logit level.** To facilitate the comparison, we first rewrite our $\ell_{\boldsymbol{\gamma},\text{ce}}$ loss in the form:

$$\ell_{\boldsymbol{\gamma},\text{ce}}(f,\mathbf{x},y) = \ln \left[ 1 + \sum_{y' \neq y} \exp\{(\mathbf{z}_{y'} - \mathbf{z}_y)/\gamma_y\} \right], \text{where } \gamma_y = \frac{\bar{c} \cdot K (\|\hat{\boldsymbol{\mu}}_y\|_2^2 + \|\hat{\mathbf{s}}_y\|_2^2)^{\frac{1}{3}}}{\sum_{k=1}^K (\|\hat{\boldsymbol{\mu}}_k\|_2^2 + \|\hat{\mathbf{s}}_k\|_2^2)^{\frac{1}{3}}} \tag{5}$$

Next, we contrast our $\ell_{\boldsymbol{\gamma},\text{ce}}$ with variants of cross-entropy that add logit margins in the unified form:

$$\ell_{\Delta,\text{ce}}(f,\mathbf{x},y) = \ln \left[ 1 + \sum_{y' \neq y} \exp\{\Delta_{yy'} + \mathbf{z}_{y'} - \mathbf{z}_y\} \right], \tag{6}$$

where $\Delta_{yy'}$ adds per-class margin into the cross-entropy loss. Typical choices for $\Delta$ include: (1) LDAM Cao et al. (2019): $\Delta_{yy'} = \mathbb{P}(y)^{-\frac{1}{4}}$ (2) EQL Tan et al. (2020): $\Delta_{yy'} = \mathbb{P}(y')$, (3) Balanced Softmax Ren et al. (2020): $\Delta_{yy'} = \ln \frac{\mathbb{P}(y')}{\mathbb{P}(y)}$, and (4) Logit Adjustment Menon et al. (2021): $\Delta_{yy'} = \tau \cdot \ln \frac{\mathbb{P}(y')}{\mathbb{P}(y)}$, with $\tau > 0$. Existing logit-margin losses are tailored to class-imbalance: their margins $\Delta_{yy'}$ are functions of skewed class priors. In contrast, our loss $\ell_{\boldsymbol{\gamma},\text{ce}}$ is derived from a non-asymptotic, class-sensitive bound (Section 4.2) and aims to improve balanced accuracy under balanced priors. Note that when the priors are equal—$\Delta_{yy'}$ becomes a constant—existing $\ell_{\Delta,\text{ce}}$ collapses to cross-entropy with *equal margins*, whereas our $\ell_{\boldsymbol{\gamma},\text{ce}}$ still imposes class-sensitive margins.

**Representation level.** The most commonly techniques used to regularize the feature representation is the contrastive objectives which take the form of

$$\ell_{\text{con}}(f,\mathbf{x},y) = \mathop{\mathbb{E}}_{\mathbf{x}^+ \in \mathcal{P}(\mathbf{x})} \ln \left[ 1 + \sum_{\mathbf{x}^- \in \mathcal{N}(\mathbf{x})} \exp\{\phi(\mathbf{x})^\top \phi(\mathbf{x}^-) - \phi(\mathbf{x})^\top \phi(\mathbf{x}^+)\} \right] \tag{7}$$

In the unsupervised setting He et al. (2020); Chen et al. (2020a), positive samples $\mathbf{x}^+ \in \mathcal{P}(\mathbf{x})$ are random augmentations of $\mathbf{x}$, while negative samples $\mathbf{x}^- \in \mathcal{N}(\mathbf{x})$ are randomly drawn from the dataset. In the supervised setting Khosla et al. (2020); Radford et al. (2021), positives share the same label as $\mathbf{x}$, and negatives come from different classes. However, these methods do not incorporate margin constraints into the objective. Some methods incorporate *fixed margins* into contrastive objectives, either tuned on a validation set Sohn (2016); Jitkrittum et al. (2022); Barbano et al. (2023) or derived from class priors for imbalanced learning Samuel & Chechik (2021). In contrast, our loss $\ell_{\bar{s}}$ adaptively sets the margin based on the average mean squared deviation based on our analysis of our generalization bound (Proposition 1).

**Discussion with Neural Collapse.** Neural Collapse (Papyan et al., 2020; Behnia et al., 2023; Kini et al., 2024; Galanti et al., 2021; Ma et al., 2025; Fang et al., 2021) characterizes idealized geometric patterns that emerge in the terminal phase of training, including the vanishing of within-class feature variance, *i.e.*, $\|\hat{\mathbf{s}}_y\| \to 0, \forall y \in \mathcal{Y}$. However, on large-scale and hard datasets such as ImageNet, this variability collapse does not materialize. Consistent with recent empirical studies, we observe that within-class feature variations remain substantial and class-dependent rather than all collapsing toward zero. These deviations motivate our modeling of non-negligible feature spreads. Detailed analysis and empirical evidence are provided in Sec.A.

**Discussion with temperature scaling.** Our logit-level loss in Eq. (5) can also be viewed as introducing a *class-dependent temperature*, similar to temperature scaling used in knowledge distillation and model calibration (Xu et al., 2020; Guo et al., 2017), where a (typically global) temperature is applied to smooth or calibrate prediction confidences. By contrast, our $\boldsymbol{\gamma}$ is *spread-aware* and is explicitly designed to balance margins between easy and hard classes, rather than improving calibration quality.

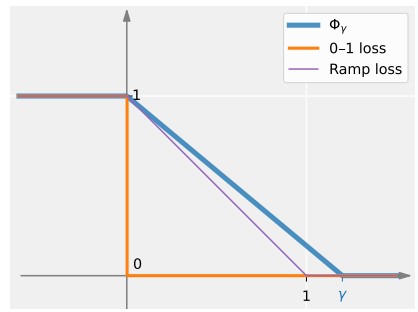

Figure 2: Plot of 0-1 loss, ramp loss and $\Phi_\gamma$.

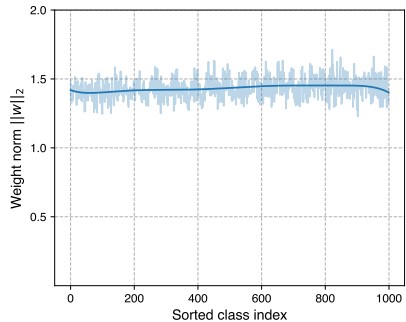

Figure 3: Weight-norm $\|\mathbf{w}_k\|_2$ on ImageNet.

## 4 THEORETICAL ANALYSIS

In this section, we explain why $\text{MR}^2$ reduces classification disparity by showing that margin regularization on the logits and representations balances the per-class error and tightens the upper bound of generalization risk. We start with some preliminaries on $\gamma$-margin loss and Rademacher complexity.

### 4.1 PRELIMINARIES

Recall that the margin of a function $f \in \mathcal{F}$ on a data point $(\mathbf{x}, y)$ is defined as:

$$\gamma_f(\mathbf{x}, y) = f(\mathbf{x}, y) - \max_{y' \neq y} f(\mathbf{x}, y'), \tag{8}$$

which measures the difference between the score assigned to the true and that of the runner-up. $\gamma$-margin loss Tian & Man-Cho So (2022) is an extension of ramp loss which is defined as follows:

**Definition 1.** (*$\gamma$-margin loss*) *Let $\Phi_\gamma(u) = \min(1, \max(0, 1 - u/\gamma))$. For any $\boldsymbol{\gamma} = [\gamma_1, ..., \gamma_K] > \mathbf{0}$, the multi-class $\gamma$-margin loss $\ell_{\boldsymbol{\gamma}} : \mathcal{F} \times \mathcal{X} \times \mathcal{Y} \to [0, 1]$ is:*

$$\ell_{\boldsymbol{\gamma}}(f, \mathbf{x}, y) = \Phi_{\gamma_y}(\gamma_f(\mathbf{x}, y)) \tag{9}$$

Figure 2 provides a comparison between $\Phi_\gamma$, the 0-1 loss, and ramp loss to better illustrate their differences. Note that the 0-1 loss is upper bounded by ramp loss and $\Phi_\gamma$. The empirical risk of $\gamma$-margin loss is defined as $\widehat{\mathcal{R}}_{\mathcal{D}}^\gamma(f) = \frac{1}{N} \sum_{\mathbf{x}, y \in \mathcal{D}} [\ell_{\boldsymbol{\gamma}}(f, \mathbf{x}, y)]$ and the expected risk is $\mathcal{R}^\gamma(f) = \mathbb{E}_{\mathbf{x}, y}[\ell_{\boldsymbol{\gamma}}(f, \mathbf{x}, y)]$. Next, we give the definition of empirical $\gamma$-margin Rademacher complexity.

**Definition 2.** (*Empirical $\gamma$-margin Rademacher complexity*) *Let $I_k$ denote the indices of the instances that belong to class $k$. Given non-negative margins $\boldsymbol{\gamma} = [\gamma_k]_{k \in [K]}$, the empirical $\gamma$-margin Rademacher complexity of $\mathcal{F}$ for a sampled dataset $\mathcal{D}$ is defined as:*

$$\widehat{\mathfrak{R}}_{\mathcal{D}}^\gamma(\mathcal{F}) = \frac{1}{N} \mathbb{E}_{\boldsymbol{\epsilon}} \left[ \sup_{f \in \mathcal{F}} \left\{ \sum_{k=1}^K \sum_{i \in I_k} \sum_{y \in \mathcal{Y}} \epsilon_{iy} \frac{f(\mathbf{x}_i, y)}{\gamma_k} \right\} \right], \tag{10}$$

*where $\boldsymbol{\epsilon} = (\epsilon_{iy})_{i,y}$ with $\epsilon_{iy}$ being i.i.d. Rademacher variables sampled uniformly from $\{-1, +1\}$.*

### 4.2 MARGIN REGULARIZATION PROMOTES GENERALIZATION AND REDUCES DISPARITY

In the sequel, we show that the margin regularization introduced in Eq.(1) and Eq.(2) improves both generalization and balanced accuracy. Our key result relies on the following lemma, which provides a bound on the expected risk in terms of the empirical $\gamma$-margin risk and its Rademacher complexity.

**Lemma 1.** (*$\gamma$-margin bound*). *Let $\mathcal{F}$ be a set of real valued functions. Given $\gamma > 0$, then for any $\delta > 0$, with probability at least $1 - \delta$, for all $f \in \mathcal{F}$:*

$$\mathcal{R}(f) \leq \widehat{\mathcal{R}}_{\mathcal{D}}^\gamma(f) + 4\sqrt{2K}\widehat{\mathfrak{R}}_{\mathcal{D}}^\gamma(\mathcal{F}) + 3\sqrt{\frac{\ln \frac{2}{\delta}}{2N}} \tag{11}$$

The proof is provided in Section B.3. This learning guarantee is a multi-class extension of Theorem 5.8 in Mohri et al. (2018), and is similar to that of Cortes et al. (2025). Furthermore, following the techniques in Theorem 5.9 of Mohri et al. (2018), Lemma 1 can be generalized to hold uniformly over all $\gamma$, with only a modest overhead of ln-ln terms. Next, we aim to **(1)** bound the empirical $\gamma$-margin Rademacher complexity $\widehat{\mathfrak{R}}_{\mathcal{D}}^{\gamma}(\mathcal{F})$ in Lemma 2, and **(2)** bound the empirical risk of $\gamma$-margin loss $\widehat{\mathcal{R}}_{\mathcal{D}}^{\gamma}$ by its cross-entropy surrogate $\widehat{\mathcal{R}}_{\mathcal{D}}^{\gamma,\text{ce}}$ in Lemma 3.

**Lemma 2.** (***Bound on the empirical Rademacher complexity***.) *For non-negative $\gamma > 0$, consider* $\mathcal{F} = \left\{ (\mathbf{x}, y) \mapsto \mathbf{w}_y^\top \phi(\mathbf{x}) \mid \mathbf{w}_y \in \mathbb{R}^d, \|\mathbf{w}_y\|_2 \leq \Lambda \right\}$. *Let* $\{\|\hat{\boldsymbol{\mu}}_k\|_2^2\}_{k=1}^K$ *and* $\{\|\hat{\mathbf{s}}_k\|_2^2\}_{k=1}^K$ *denote the per-class squared norms of the feature means and the mean squared deviations, respectively, as defined in Section 3. Then, the following bound holds for all $f \in \mathcal{F}$:*

$$\widehat{\mathfrak{R}}_{\mathcal{D}}^{\gamma}(\mathcal{F}) \leq \Lambda \sqrt{\frac{K}{N}} \sqrt{\sum_{k=1}^K \frac{\|\hat{\boldsymbol{\mu}}_k\|_2^2 + \|\hat{\mathbf{s}}_k\|_2^2}{\gamma_k^2}} \tag{12}$$

The proof is provided in Section B.4. This bound is **class-sensitive**, as it depends on the per-class squared norms of the means, per-class mean squared deviations and the class-specific margins, *i.e.*, $\|\hat{\boldsymbol{\mu}}_k\|_2^2$, $\|\hat{\mathbf{s}}_k\|_2^2$ and $\gamma_k^2$. Note that we use a uniform bound $\Lambda$ on the $L_2$ norms of the classification head weight $\|\mathbf{w}\|_2$ rather than class-wise bounds, motivated by the key finding of Cui et al. (2024): disparity does not stem from classifier bias, and the $L_2$ norms of the weights $\|\mathbf{w}\|_2$ are already balanced, as illustrated in Figure 3, which shows $\|\mathbf{w}\|_2$ values from a ResNet-50 trained on ImageNet. In training, we use the cross-entropy surrogate $\ell_{\gamma,\text{ce}}$ in place of the $\gamma$-margin loss $\ell_\gamma$. Let $\widehat{\mathcal{R}}_{\mathcal{D}}^{\gamma,\text{ce}}(f) = \frac{1}{N} \sum_{\mathbf{x},y \in \mathcal{D}} [\ell_{\gamma,\text{ce}}(f, \mathbf{x}, y)]$. The following Lemma relates $\widehat{\mathcal{R}}_{\mathcal{D}}^{\gamma}$ and $\widehat{\mathcal{R}}_{\mathcal{D}}^{\gamma,\text{ce}}$.

**Lemma 3.** *For $\gamma > 0$. Then, $\forall f \in \mathcal{F}$, we have $\widehat{\mathcal{R}}_{\mathcal{D}}^{\gamma}(f) \leq \frac{1}{\ln 2} \widehat{\mathcal{R}}_{\mathcal{D}}^{\gamma,\text{ce}}(f)$.*

The proof is provided in Section B.5. Building on Lemmas 1, 2 and 3, we derive the following class-sensitive learning guarantee for margin-based cross-entropy.

**Proposition 1.** *For $\gamma > 0$, consider* $\mathcal{F} = \left\{ (\mathbf{x}, y) \mapsto \mathbf{w}_y^\top \phi(\mathbf{x}) \mid \mathbf{w}_y \in \mathbb{R}^d, \|\mathbf{w}\|_2 \leq \Lambda \right\}$. *Let* $\{\|\hat{\boldsymbol{\mu}}\|_2^2\}_{k=1}^K$ *and* $\{\|\hat{\mathbf{s}}\|_2^2\}_{k=1}^K$ *denote the per-class squared norms of the feature means and the mean squared deviations, respectively. Then, the following bound holds for all $f \in \mathcal{F}$:*

$$\mathcal{R}(f) \leq \frac{1}{\ln 2} \widehat{\mathcal{R}}_{\mathcal{D}}^{\gamma,\text{ce}}(f) + \frac{4\sqrt{2}\Lambda K}{\sqrt{N}} \sqrt{\sum_{k=1}^K \frac{\|\hat{\boldsymbol{\mu}}_k\|_2^2 + \|\hat{\mathbf{s}}_k\|_2^2}{\gamma_k^2}} + \mathcal{O}(1/\sqrt{N}). \tag{13}$$

This bound implies that if $f$ maintains a low empirical risk $\widehat{\mathcal{R}}_{\mathcal{D}}^{\gamma,\text{ce}}(f)$ —often achievable with over-parameterized deep nets in practice — with relatively large margin values $\gamma_k$, then it enjoys a lower generation risk, as reflected by the second (complexity) term. However, overly large $\gamma_k$ is undesirable. Because $\ell_{\gamma,\text{ce}}$ grows monotonically with $\gamma$, overly increasing $\gamma_k$ inflates the empirical margin risk and makes the loss harder to optimize. Let the average margin budget be fixed as $\bar{c} = \frac{1}{K} \sum_k \gamma_k$. Under this constraint, our choice of $\gamma$ is determined below.

**Corollary 1.** (***Our choice of $\gamma$***) *For fixed $\bar{c}$, the complexity term of the bound is minimized when*

$$\gamma_y = \frac{\bar{c} K (\|\hat{\boldsymbol{\mu}}_y\|_2^2 + \|\hat{\mathbf{s}}_y\|_2^2)^{\frac{1}{3}}}{\sum_{k=1}^K (\|\hat{\boldsymbol{\mu}}_k\|_2^2 + \|\hat{\mathbf{s}}_k\|_2^2)^{\frac{1}{3}}} \quad \text{for all } y \in [K]. \tag{14}$$

This follows from solving a Lagrangian optimization problem, with the proof provided in Section B.6.

**Remark.** Analogous to Eq. 13, we bound the per-class error $\mathcal{R}_k$ for class $k$ w.p. at least $1 - \delta$:

$$\mathcal{R}_k(f) \leq \frac{1}{\ln 2} \widehat{\mathcal{R}}_{\mathcal{D}_k}^{\gamma,\text{ce}} + \frac{4}{\gamma_k} \widehat{\mathfrak{R}}_k(\mathcal{F}) + \varepsilon(\delta, N), \tag{15}$$

where $\widehat{\mathfrak{R}}_k \propto \sqrt{\|\hat{\boldsymbol{\mu}}_k\|_2^2 + \|\hat{\mathbf{s}}_k\|_2^2}$ and $\epsilon$ is a low-order term (see proof in Section B.8).

Corollary 1 motivates our **logit-level margin regularizer**: choosing $\gamma_k \propto (\|\hat{\boldsymbol{\mu}}_k\|_2^2 + \|\hat{\mathbf{s}}_k\|_2^2)^{1/3}$ minimizes the generalization bound and balances Eq. (15), since the feature variability term

Table 1: **Comparison against existing methods** in reducing class-wise accuracy disparity on CIFAR-100 and ImageNet. ResNet-32 and CLIP ResNet-50 backbones are adopted respectively.

|  (a) **CIFAR-100** | | | | | (b) **ImageNet** | | | |
| --- | --- | --- | --- | --- | --- | --- | --- | --- |
| Method | Overall | Easy | Medium | Hard | Method | Overall | Easy | Medium | Hard |
| ERM | 70.9 | 84.5 | 71.0 | 56.7 | ERM | 75.2 | 91.1 | 78.3 | 56.4 |
| LfF | 69.1 | 83.6 (-0.9) | 70.1 (-0.9) | 53.7 (-3.0) | LfF | 74.4 | 90.2 (-0.9) | 77.8 (-0.5) | 55.2 (-1.2) |
| JTT | 70.6 | 84.3 (-0.2) | 70.8 (-0.2) | 56.2 (-0.5) | JTT | 74.8 | 90.7 (-0.4) | 77.9 (-0.4) | 55.7 (-0.4) |
| EQL | 70.7 | 84.4 (-0.1) | 70.9 (-0.1) | 56.4 (-0.3) | EQL | 75.3 | 91.3 (+0.2) | 78.4 (+0.3) | 56.2 (-0.2) |
| LDAM | 71.1 | 84.7 (+0.2) | 71.2 (+0.2) | 57.0 (+0.3) | LDAM | 75.4 | 91.5 (+0.4) | 78.6 (+0.3) | 56.1 (-0.3) |
| LGM | 70.8 | 84.0 (-0.5) | 71.4 (+0.4) | 56.6 (-0.1) | LGM | 75.3 | 90.9 (-0.2) | 78.3 (+0.0) | 56.6 (+0.2) |
| CMIC-DL | 71.1 | 85.1 (+0.6) | 70.9 (-0.1) | 56.8 (+0.1) | CMIC-DL | 75.2 | 91.0 (-0.1) | 78.2 (-0.1) | 56.5 (+0.1) |
| SAM | 71.0 | 84.6 (+0.1) | 71.1 (+0.1) | 56.9 (+0.2) | SAM | 75.6 | 91.4 (+0.3) | 78.8 (+0.5) | 56.6 (+0.2) |
| DFL | 71.3 | 84.8 (+0.3) | 71.3 (+0.3) | 57.1 (+0.4) | DFL | 75.8 | 91.7 (+0.6) | 78.6 (+0.3) | 57.1 (+0.7) |
| SupCon | 71.5 | 85.0 (+0.5) | 72.1 (+1.1) | 57.5 (+0.8) | SupCon | 75.7 | 91.4 (+0.3) | 78.9 (+0.6) | 56.7 (+0.3) |
| DRL | 71.9 | 85.5 (+1.0) | 72.5 (+1.5) | 57.7 (+1.0) | DRL | 75.5 | 91.4 (+0.3) | 78.5 (+0.2) | 56.6 (+0.2) |
| CSR | 71.2 | 85.1 (+0.6) | 71.3 (+0.3) | 57.1 (+0.4) | CSR | 75.1 | 90.8 (-0.3) | 77.9 (-0.4) | 56.6 (+0.2) |
| DRO | 71.6 | 85.1 (+0.6) | 72.2 (+1.2) | 57.2 (+0.5) | DRO | 75.9 | 91.0 (-0.1) | 79.3 (+1.0) | 57.2 (+0.8) |
| FairDRO | 72.0 | 85.7 (+1.2) | 72.5 (+1.5) | 57.7 (+1.0) | FairDRO | 76.1 | 91.5 (+0.4) | 79.6 (+1.3) | 57.3 (+0.9) |
| MR$^2$ (ours) | **73.9** | **85.9** (+1.4) | **73.8** (+2.8) | **61.9** (+5.2) | MR$^2$ (ours) | 76.9 | 91.5 (+0.4) | 79.7 (+1.4) | 59.6 (+3.2) |

($\sqrt{\|\hat{\boldsymbol{\mu}}_k\|_2^2 + \|\hat{\mathbf{s}}_k\|_2^2}$) appears in the numerator while $\gamma_k$ controls the denominator. Their proportionality yields a more balanced contribution across classes.

Proposition 1 justifies our **representation-level regularizer**: the loss $\ell_{\bar{s}}$ reduces the mean-squared deviations $\|\hat{\mathbf{s}}_k\|_2^2$, thereby tightening the complexity (second) term in Eq. (13). By suppressing large variances in harder classes, this regularization also balances the per-class error bounds, preventing them from being dominated by high-variance classes.

### 4.3 GENERAL NORM BOUNDS ON MARGIN-BASED COMPLEXITY

Astute readers may wonder whether Proposition 1 remains informative for cosine classifier models, *e.g.*, CLIP models (Radford et al., 2021), where features are $L_2$-normalized, *i.e.*, $\|\phi(\mathbf{x})\|_2 = 1$. In this case, the factor $\|\hat{\boldsymbol{\mu}}_y\|_2^2 + \|\hat{\mathbf{s}}_y\|_2^2 = \frac{1}{|I_k|}\sum_{i\in I_k}\|\phi(\mathbf{x}_i)\|_2^2 = 1$ in the complexity term becomes constant, making it class-agnostic. Fortunately, Proposition 1 naturally extends to other norm-based formulations, as in Proposition 2. This allows margin regularization under any $L_p$ norm with $p \geq 1$.

**Proposition 2 (General bound on the $\gamma$-margin Rademacher complexity).** *Fix conjugate exponents $p, q \in [1, \infty]$ such that $1/p + 1/q = 1$. Let $\mathcal{F}_q = \{(x, y) \mapsto \mathbf{w}_y^\top \phi(\mathbf{x}) \mid \mathbf{w}_y \in \mathbb{R}^d, \|\mathbf{w}_y\|_q \leq \Lambda_q\}$. For each class $k \in [K]$, we write the average $L_p^2$ norm as $r_{k,p}^2 = \frac{1}{|I_k|}\sum_{i\in I_k}\|\phi(\mathbf{x}_i)\|_p^2$. Then, for all $f \in \mathcal{F}_q$, the empirical $\gamma$-margin Rademacher complexity satisfies*

$$\widehat{\mathfrak{R}}_{\mathcal{D}}^{\gamma}(\mathcal{F}_q) \leq C(p)\Lambda_q\sqrt{\frac{K}{N}}\sqrt{\sum_{k=1}^{K}\frac{r_{k,p}^2}{\gamma_k^2}} \tag{16}$$

*where $C(p)$ is a constant factor depends on $p$ (its exact expression and proof appear in Section B.7).*

Analogously to Corollary 1, the corresponding choice of $\gamma$ under $L_p$ norm with $p \in [1, \infty]$ is

$$\gamma_{y,p} = \frac{\bar{c}Kr_{y,p}^{2/3}}{\sum_k r_{k,p}^{2/3}}. \tag{17}$$

For cosine classifiers, one can restore class-dependent margin by selecting non-$L_2$ norms, *e.g.*, $p = 3$.

## 5 EXPERIMENTS

**Datasets and backbones.** We select seven datasets: CIFAR-100 (Krizhevsky & Hinton, 2009), ImageNet (Deng et al., 2009), StanfordCars (Krause et al., 2013), OxfordPets (Parkhi et al., 2012), Flowers (Nilsback & Zisserman, 2008), Food (Bossard et al., 2014), and FGVCAircraft (Maji et al., 2013), using both CNNs (He et al., 2016a;b) and Vision Transformers (Dosovitskiy et al., 2021).

Table 2: **Evaluation across diverse model architecture.** We perform **linear probing** on MoCov2 ResNet-50 using only $\ell_{\gamma,\text{ce}}$. CLIP models are **cosine classifier** models. RN: ResNet. W-RN: WideResNet. TfS: Train from Scratch.

|  | (a) **CIFAR-100** | | | |  | (b) **ImageNet** | | | |
| --- | --- | --- | --- | --- | --- | --- | --- | --- | --- |
| Model | Overall | Easy | Medium | Hard | Model | Overall | Easy | Medium | Hard |
| ResNet-20 | 68.7 | 83.3 | 70.3 | 53.0 | RN-50 TfS | 71.7 | 88.5 | 74.1 | 52.6 |
| + MR$^2$ | 70.9 | 84.4 (+1.1) | 72.2 (+1.9) | 56.7 (+3.7) | + MR$^2$ | 74.2 | 89.9 (+1.4) | 76.9 (+2.8) | 55.9 (+3.3) |
| PreAct RN-20 | 69.1 | 83.8 | 70.5 | 53.4 | MoCov2 RN-50 | 71.1 | 89 | 73.5 | 50.7 |
| + MR$^2$ | 71.3 | 84.7 (+0.9) | 73.2 (+2.7) | 56.6 (+3.2) | + MR$^2$ ($\ell_{\gamma,\text{ce}}$) | 72.4 | 89.6 (+0.6) | 74.8 (+1.3) | 52.7 (+2.0) |
| PreAct RN-32 | 71.2 | 85.0 | 71.7 | 56.7 | CLIP ViT-B/32 | 75.6 | 91.4 | 78.4 | 57.0 |
| + MR$^2$ | 73.3 | 86.1 (+1.1) | 73.9 (+2.2) | 60.1 (+3.4) | + MR$^2$ | 77.1 | 91.7 (+0.3) | 79.9 (+1.5) | 59.8 (+2.8) |
| W-RN-22-10 | 78.4 | 89.6 | 80.0 | 65.4 | MAE ViT-B/16 | 80.4 | 94.5 | 83.9 | 62.7 |
| + MR$^2$ | 81.2 | 90.7 (+1.1) | 82.5 (+2.5) | 70.2 (+4.8) | + MR$^2$ | 82.0 | 94.8 (+0.3) | 85.3 (+1.4) | 66.1 (+3.4) |

For CIFAR-100, we use ResNet-{20, 32} (He et al., 2016a) , PreAct ResNet-{20, 32} (He et al., 2016b), and WideResNet-22-10 (Zagoruyko & Komodakis, 2016) as our backbones. For training from scratch on ImageNet, we use ResNet-50 (He et al., 2016a) as the backbone. For fine-tuning on ImageNet, we adopt four pre-trained models: MAE (He et al., 2022) ViT-B/16, MoCov2 (Chen et al., 2020b) ResNet-50, and CLIP (Radford et al., 2021) with ResNet-50 and ViT-B/32. For the rest datasets, we use CLIP ResNet-50.

**Baselines.** We compare our MR$^2$ against 14 baselines: (1) ERM (Vapnik, 1991), empirical risk minimization with cross-entropy loss. Error-based reweighting methods: (2) LfF (Nam et al., 2020) and (3) JTT (Liu et al., 2021). Margin adjustment methods: (4) EQL (Tan et al., 2020) and (5) LDAM (Cao et al., 2019). Contrastive representation learning methods: (6) SupCon (Khosla et al., 2020) and (7) DRL (Samuel & Chechik, 2021). (8) CSR (Jin et al., 2025), a regularization technique for enhancing robust fairness. Distributionally robust optimization methods: (9) DRO (Sagawa et al., 2020) and (10) FairDRO (Jung et al., 2023). (11) Sharpness-Aware Minimization (SAM) (Foret et al., 2021). Methods that promote class separability: (12) DFL (Wen et al., 2016) (13) LGM (Wan et al., 2018) (14) CMIC-DL (Yang et al., 2025) Further details of the baselines are given in Section C.1.

**Evaluation metrics.** We report the average top-1 accuracy as the overall performance metric. To assess performance gap across classes, we follow Cui et al. (2024) to partition each dataset into three equal-sized subsets—"easy", "medium", and "hard"—based on per-class accuracy ranking. For instance, classes in the "easy" group correspond to the top one-third in per-class accuracy. We report the average accuracy for each of the three subsets. In Section C.3, we also report the per-class accuracies of MR$^2$ to without the partition.

**Implementation details.** For CIFAR-100, our training configurations are aligned with Cao et al. (2019). We follow the training setup of He et al. (2022) for fine-tuning MAE and He et al. (2020) for MoCov2, and adopt the configuration from Wortsman et al. (2022) for CLIP ViT-B/32 and ResNet-50. The decay rate for exponential moving average (EMA) is fixed at 0.9 across all experiments. The hyper-parameter $\bar{c}$ in Eq.(1) is selected from $\{1, 2, 3\}$ and $\lambda$ in Eq.(4) is tuned over $\{0.1, 0.3, 0.5, 0.7, 0.9\}$ on validation sets. In the case of cosine classifier models (*e.g.*, CLIP), we adopt $p = 3$ in Eq. (17). By default, we use standard augmentations such as random cropping and flipping for fair comparison. We further evaluate our MR$^2$ under advanced augmentation strategies (*e.g.*, RandAugment(Cubuk et al., 2020) and AutoAug (Cubuk et al., 2019)); the results appear in Table 5 Sec. A. All experiments are repeated three times, standard deviations are reported in Tab. 15. See Section C.2 for more details.

## 5.1 MAIN RESULTS

**Comparison against existing methods.** We select ResNet-32 and CLIP ResNet-50 as backbones for CIFAR-100 and ImageNet, respectively, and compare the results with various baselines in Table 1. We observe that: (1) Reweighting methods (LfF, JTT) underperform ERM, often degrading performance on "hard" classes. (2) Margin adjustment methods (EQL, LDAM) yield minimal gains and fail to reduce class-level disparity. (3) Representation learning and robust optimization methods (SupCon, DRL, CSR, DRO, FairDRO) improve overall accuracy but fail to reduce disparity as the gain on ges"hard" classes is even lower than that on "medium/easy" classes (*e.g.*, FairDRO and SupCon)

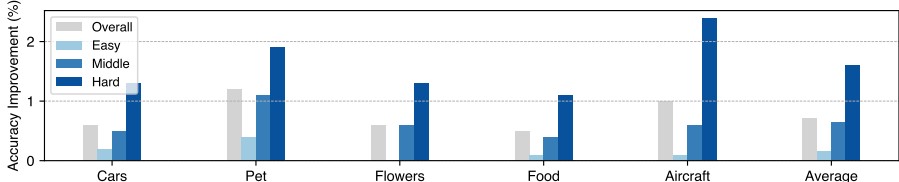

Figure 4: **Relative improvements on fine-grained datasets**: StanfordCars, OxfordPets, Flowers, Food and FGVCAircraft with CLIP ResNet-50. Numerical values are provided in Table 10.

Table 3: **Main component analysis** on CIFAR-100 using ResNet-32.

|  | Model | Overall | Easy | Medium | Hard |
|---|---|---|---|---|---|
| **(a)** | Baseline ($\ell_{\text{ce}}$) | 70.9 | 84.5 | 71.0 | 56.7 |
| **(b)** | $\ell_{\gamma,\text{ce}}$ with uniform $\gamma$ | 70.6 (-0.3) | 84.6 (+0.1) | 70.8 (-0.2) | 56.5 (-0.2) |
| **(c)** | $\ell_{\gamma,\text{ce}}$ | 72.8 (+1.9) | 85.1 (+0.6) | 72.9 (+1.9) | 60.4 (+3.7) |
| **(d)** | $\ell_{\text{ce}} + \lambda\ell_{\bar{s}}$ with $\bar{s}=0$ | 71.6 (+0.7) | 85.0 (+0.5) | 72.2 (+1.2) | 57.6 (+0.9) |
| **(e)** | $\ell_{\text{ce}} + \lambda\ell_{\bar{s}}$ | 72.3 (+1.4) | 85.4 (+0.9) | 73.1 (+2.1) | 58.8 (+2.1) |
| **(f)** | $\text{MR}^2$: $\ell_{\gamma,\text{ce}} + \lambda\ell_{\bar{s}}$ | 73.9 (+3.0) | 85.9 (+1.4) | 73.8 (+2.8) | 61.9 (+5.2) |

and sometimes exhibit trade-offs, *e.g.*, on ImageNet, DRO improves "hard" classes by $0.8\%$ while sacrificing "easy" classes by $-0.1\%$. (4) Our $\text{MR}^2$ achieves consistent improvements across both datasets, substantially boosting "hard" class accuracy while mildly improving on "easy" one, thus reducing the performance disparity without trade-offs.

**Evaluation across diverse model architectures.** Table 2 presents all metrics on CIFAR-100 and ImageNet. For MoCov2, we follow its standard evaluation protocol to perform linear classification on frozen features by omitting the representation margin loss $\ell_{\bar{s}}$. Recall that CLIP models employ cosine classifiers, and we set $p = 3$ in Eq. (17). Our $\text{MR}^2$ achieve consistent improvements across all metrics, indicating improved generalization. Notably, our method effectively reduces class-wise performance disparity, with substantially larger gains on "hard" classes than those on "easy" classes. Our evaluation across linear classifiers, cosine classifier models, training from scratch, diverse pretraining objectives and model architectures validates the general effectiveness of $\text{MR}^2$.

**Evaluation across fine-grained datasets.** Figure 4 illustrates the relative performance gains of $\text{MR}^2$ on five fine-grained benchmarks using CLIP ResNet-50, reinforcing its benefit for both balanced performance and generalization. Complete results for all metrics are in Table 10 in Section C.3.

## 5.2 ABLATION STUDIES

**Main component analysis.** In Table 3, we ablate the effect of our margin regularization on logits and feature embeddings on CIFAR-100 using ResNet-32. Row **(a)** is the baseline trained with cross-entropy loss; Rows **(b)** and **(c)** apply our logit margin loss $\ell_{\gamma,\text{ce}}$, where Row **(b)** uses a simple uniform margin tuned on validation set ($\gamma_k = \bar{c}, \forall k \in [K]$), and Row **(c)** uses our proposed class-wise margins; Rows **(d)** and **(e)** apply our representation margin loss $\ell_{\bar{s}}$ on top of standard cross-entropy loss $\ell_{\text{ce}}$, where Row **(d)** sets $\bar{s} = 0$ (removing feature-level margins), and Row **(e)** uses our full $\ell_{\bar{s}}$ formulation. Row **(f)** is our $\text{MR}^2$, combining both $\ell_{\gamma,\text{ce}}$ and $\ell_{\bar{s}}$. Table 11 in Sec. C.3 provides further results on ImageNet and CIFAR-100 across diverse backbones.

From Rows **(a,b,c)**, we observe that: (1) uniform margins does not improve balanced results or generalization—Row **(b)** performs similarly to Row **(a)**; and (2) our class-wise margin design in Eq. (1) leads to clear improvements on "hard" classes and moderate gains on "easy" classes. From Rows **(a,d,e)**, we find that our representation margin loss yields notable improvements across all three subsets. In contrast, setting the representation margin to $\bar{s} = 0$ fails to reduce disparity, as the gain on "hard" classes is even lower than that on "medium" classes. Finally, Row **(f)** shows that combining logit and representation margin losses further enhances both fairness and overall performance.

**Effect of $\bar{c}$.** In Figure 5, we study the impact of $\bar{c}$ in Eq. (1) by varying its value from 0 to 5. Using ResNet-32 on CIFAR-100, we plot the average accuracy for both overall and "hard" classes. A

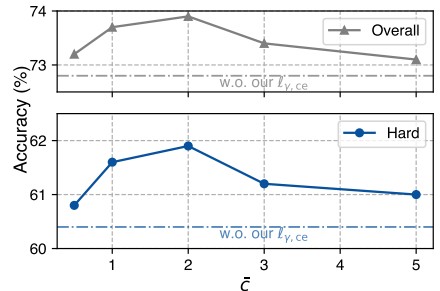

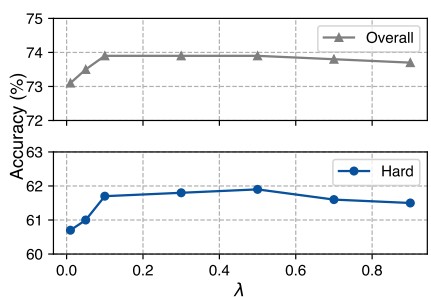

Figure 5: Effect of $\bar{c}$.

Figure 6: Effect of $\lambda$.

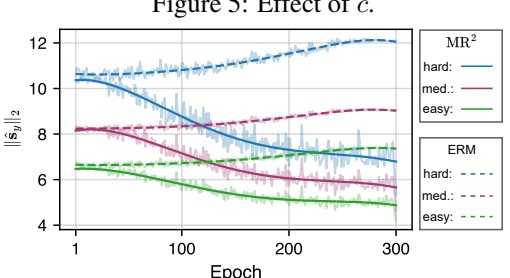

Figure 7: Visualization of the evolving $\|\hat{\mathbf{s}}_y\|$.

Table 4: Effect of $\mathrm{MR}^2$ on output-level and classifier-level margins. (↑): higher is better. C.-level: classifier-level margin. (Formal definition of the two margins in text)

| | Output-level (↑) | | | | C.-level (↑) |
|---|---|---|---|---|---|
| | Avg. | Easy | Med. | Hard | |
| ERM | 0.158 | 0.236 | 0.144 | 0.093 | 58.4 |
| $\mathrm{MR}^2$ | 0.373 | 0.435 | 0.396 | 0.289 | 81.8 |

small $\bar{c}$ leads to insufficient regularization, while a large $\bar{c}$ imposes overly strong constraints, making optimization difficult. Both curves exhibit similar trends, with optima located around $\bar{c} = 2$.

**Effect of $\lambda$.** In Figure 6, we examine the effect of $\lambda$ in Eq.(4) by varying its value from 0 to 0.9 and reporting the accuracy of overall and "hard" classes on CIFAR-100 using ResNet-32. $\mathrm{MR}^2$ achieves optimal overall and "hard" classes performance around $\lambda = 0.5$, and the performance remains stable in this region, indicating low sensitivity to $\lambda$. See Figure 8-9 for the study of $\bar{c}$ and $\lambda$ on ImageNet.

**Using different norms $p$.** By default, we use $L_2$ norm for all experiments except for $L_3$ norm when applied to cosine classifiers. Table 8 and Table 9 (deferred to Appendix C.3 due to space limits) report the results with various $L_p$ norms on CIFAR-100 and ImageNet. As expected, using $p \geq 1$ consistently improves both fairness and generalization, supporting our theoretical framework.

**Visualization of the evolution of $\|\hat{\mathbf{s}}_y\|_2$.** Figure 7 illustrates the evolution of the average $\|\hat{\mathbf{s}}_y\|_2$ for easy, medium, and hard classes when training ResNet-50 from scratch on ImageNet. The contrast is clear: $\mathrm{MR}^2$ consistently compresses the gap between easy and hard classes throughout training, whereas the baseline exhibits a steadily widening disparity.

**Effect of $\mathrm{MR}^2$ on output-level and classifier-level margins.** Comparing logit margins across models is difficult because their logit scales may differ. We therefore evaluate the output-level margin using prediction probabilities, defined as the gap between the ground-truth probability and the runner-up one: $m_{\mathrm{o}}(\mathbf{x}, y) = \mathbb{P}_\theta(y \mid \mathbf{x}) - \max_{y' \neq y} \mathbb{P}_\theta(y' \mid \mathbf{x})$. For classifier-level margin, we adopt the minimal pairwise angle distance among the classifier weights $\{\mathbf{w}_k\}_{k=1}^K$ from Zhou et al. (2022): $m_{\mathrm{c}}(\{\mathbf{w}_k\}_{k=1}^K) = \min_{i \neq j} \angle(\mathbf{w}_i, \mathbf{w}_j) = \arccos[\max_{i \neq j} \mathbf{w}_i^\top \mathbf{w}_j/(\|\mathbf{w}_i\|_2\|\mathbf{w}_j\|_2)]$. As in Table 4, $\mathrm{MR}^2$ significantly enlarges output-level margins across all subsets, reduces the margin gap between easy and hard class, and raises the classifier-level margin from 58.4 to 81.8.

## 6 CONCLUSION

We present $\mathrm{MR}^2$, a theoretically grounded framework that reduces class-wise performance gap by regularizing both logit and representation margins. Built upon a novel class-sensitive learning guarantee, our method links generalization error to per-class feature variability and margin design. Guided by this insight, $\mathrm{MR}^2$ assigns adaptive margins to balance generalization bounds across classes and promotes intra-class compactness. Extensive experiments across seven datasets, diverse architectures, and pre-trained foundation models demonstrate that $\mathrm{MR}^2$ consistently improves performance on "hard" classes without degrading accuracy on "easy" ones, enhancing both fairness and overall performance.

ACKNOWLEDGMENT

This project is partially supported by the Ministry of Education, Singapore, under its Tier-1 Academic Research Fund (No. 24-SIS-SMU-040). This research is also supported by the RIE2025 Industry Alignment Fund – Industry Collaboration Projects (IAF-ICP)(Award I2301E0026), administered by A*STAR, as well as supported by Alibaba Group and NTU Singapore through Alibaba-NTU Global e-Sustainability CorpLab (ANGEL).

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

CONTENTS

## A  ADDITIONAL DISCUSSIONS

***Discussion 1:*** *Would it be better to make $\gamma$ trainable after initialization according to Eq. (1) ?*

We do not find this beneficial. Since our loss $\ell_{\gamma,\text{ce}}$ is monotonically increasing in $\gamma$, treating $\gamma$ as a trainable parameter would introduce a trivial descent direction: the optimizer can always reduce the loss by shrinking $\gamma$. As a result, $\gamma$ would systematically converge toward small values, collapsing the class-dependent margins and failing to enforce a balanced margin across easy and hard classes.

Table 5: $\text{MR}^2$ with advanced data augmentations on ImageNet using CLIP-ResNet-50.

| Model | Overall | Easy | Medium | Hard |
|---|---|---|---|---|
| Standard data augmentation | 75.2 | 91.1 | 78.3 | 56.4 |
| + $\text{MR}^2$ | 76.9 (+1.7) | 91.5 (+0.4) | 79.7 (+1.4) | 59.6 (+3.2) |
| RangAug | 75.8 | 91.5 | 78.8 | 57.2 |
| + $\text{MR}^2$ | 77.3 (+1.5) | 92.0 (+0.5) | 79.8 (+1.0) | 60.1 (+2.9) |
| AutoAug | 76.2 | 91.9 | 79.2 | 57.5 |
| + $\text{MR}^2$ | 78.1 (+1.9) | 92.3 (+0.4) | 81.0 (+1.8) | 61.1 (+3.5) |

***Discussion 2:*** *Effectiveness of our $\text{MR}^2$ with advanced data augmentations.*

In our main experiments and ablation studies, we adopt standard data augmentations, *i.e.*, random cropping and flipping. Here, we further evaluate the effectiveness of $\text{MR}^2$ under stronger data augmentation strategies. Table5 reports results using RandAug (Cubuk et al., 2020) and AutoAug (Cubuk et al., 2019) on ImageNet with CLIP-ResNet-50. Again, $\text{MR}^2$ consistently boosts the accuracy of "hard" classes while providing mild improvements for easy ones, thereby reducing performance disparity without introducing trade-offs.

***Discussion 3:*** *Relation to Neural Collapse.*

Neural Collapse (NC, (Papyan et al., 2020)) provides a useful lens for understanding the geometry of deep classifiers in the terminal phase of training (TPT). Two empirical properties are particularly relevant to our setting:

- **(NC1) Classifier collapse to an ETF structure.** The classifier weights converge to equal-norm vectors whose pairwise angles are all equal, forming an equiangular tight frame (ETF).
- **(NC2) Variability collapse of last-layer features.** Within-class feature variance collapses and features concentrate tightly around their respective class means.

In our experiments, we observe that these idealized NC patterns do not fully emerge on large-scale datasets, such as ImageNet. Specifically, equal norms in NC1 are approximately realized (which is consistent with our theoretical assumption of a uniform $L_2$-norm bound $\Lambda$ on the classifier weights, rather than a class-dependent one), but the ETF structure of equal pairwise angles and the full variability collapse in NC2 are violated under our settings.

Our observations are consistent with the empirical study of Cui et al. (2024), who adapt NC diagnostic tools to measure feature and classifier variability. Their results show that (i) although classifier weights become approximately equal-norm, their pairwise angles do not converge to an ETF configuration—"easy" classes exhibit larger separation angles (Sec. 4.1 and Fig. 5 in Cui et al. (2024)); and (ii) the within-class feature variations remain non-negligible, with an average ratio between variance norm and mean feature norm of $\text{Avg}_y[\|\hat{\mathbf{s}}_y\|/\|\hat{\boldsymbol{\mu}}_y\|] \approx 0.3285$, and these variations are imbalanced between easy and hard classes (Sec. 4.3 and Fig. 2 in Cui et al. (2024)).

The gap between the ideal NC behavior and the empirical geometry observed on datasets such as ImageNet can be attributed to at least two factors. First, **task hardness**. The difficulty of a classification problem depends on the number of classes, the sample size, and the intrinsic complexity of the visual concepts. Inspecting Fig. 6 (first row) in Papyan et al. (2020), one observes that as task difficulty increases, the within-class feature variation ceases to collapse toward zero: the magnitude of the within-class to between-class variance is on the order of $10^{-4}$ for the easiest task (MNIST), but exceeds $1$ for the most challenging subsampled ImageNet setting. To further

Table 6: Per-class $\|\hat{\mathbf{s}}_y\|_2$ and $\|\hat{\boldsymbol{\mu}}_y\|_2$ on MNIST. We adopt the same training configuration as in ImageNet using ResNet-18. On MNIST, the variability collapse of features (NC2) clearly emerges, with $\mathrm{Avg}_y[\|\hat{\mathbf{s}}_y\|_2/\|\hat{\boldsymbol{\mu}}_y\|_2] \approx 7.5 \times 10^{-4}$. In contrast, this phenomenon does not occur on ImageNet, where $\mathrm{Avg}_y[\|\hat{\mathbf{s}}_y\|_2/\|\hat{\boldsymbol{\mu}}_y\|_2] \approx 0.3285$.

| class id | 1 | 2 | 3 | 4 | 5 | 6 | 7 | 8 | 9 | 10 |
|---|---|---|---|---|---|---|---|---|---|---|
| $\|\hat{\mathbf{s}}_y\|_2$ | 0.0041 | 0.0032 | 0.0048 | 0.0051 | 0.0056 | 0.0063 | 0.0040 | 0.0056 | 0.0056 | 0.0063 |
| $\|\hat{\boldsymbol{\mu}}_y\|_2$ | 6.6982 | 6.7474 | 6.5227 | 6.8819 | 6.7275 | 6.9866 | 6.4091 | 6.4091 | 6.7820 | 6.6932 |

illustrate that NC2 tends to appear only on simpler datasets, we repeat the same analysis on MNIST, using the same training configuration as in ImageNet but with a ResNet-18 backbone. The results appear in Table 6: On MNIST, the variability collapse of features (NC2) does emerge: the ratio $\mathrm{Avg}_y[\|\hat{\mathbf{s}}_y\|_2/\|\hat{\boldsymbol{\mu}}_y\|_2]$ is approximately $7.5 \times 10^{-4}$. However, recall that on ImageNet this ratio is much larger, $\mathrm{Avg}_y[\|\hat{\mathbf{s}}_y\|_2/\|\hat{\boldsymbol{\mu}}_y\|_2] \approx 0.3285$, indicating that NC2 does not hold in this harder, large-scale setting. Moreover, Papyan et al. (2020) use only 600 training samples per class—roughly half of the full ImageNet training size—yet still observe substantial residual within-class variability. This suggests that non-vanishing $\|\hat{\mathbf{s}}_y\|$ is expected for large-scale, high-complexity datasets.

Second, **practical regularization**. Modern training pipelines routinely employ regularization strategies that were not considered in Papyan et al. (2020) and that hinder complete variability collapse. Early stopping prevents networks from entering the extremely long TPT required for NC to fully emerge, since training is typically stopped according to validation performance rather than continued for many additional epochs. In addition, strong data augmentation substantially increases the effective difficulty of the task by injecting extra intra-class variation, while Papyan et al. (2020) train without augmentation.

Together, these factors naturally inhibit complete within-class variability collapse in contemporary large-scale settings, and justify modeling and regularizing the residual, class-dependent spreads $\|\hat{\mathbf{s}}_y\|$ as we do in this work.

***Discussion 4:*** *Applying our* MR$^2$ *to class-imbalanced settings.*

In the main experiments, we omit evaluation under class-imbalance settings, because our goal is to disentangle two fundamentally different sources of class-wise performance disparity: (1) imbalanced class numbers (2) imbalanced class feature spreads (margins). Evaluating on imbalanced datasets would entangle these effects and obscure the specific phenomenon we aim to study.

For class-imbalance scenarios, extensive prior work already provides well-established solutions. These methods typically attribute class-wise disparity to shifted decision boundaries and thus introduce classifier-level corrections (often termed decoupled learning in long-tail recognition), such as freezing the backbone and retraining or reweighting classifier weights (Kang et al., 2020), or adjusting classifier biases/margins (Menon et al., 2021). Notably, Menon et al. (2021) show that a Fisher-consistent solution for balanced error is achieved by subtracting the log class prior from the predicted logits (the logit-adjustment method, LA).

These classifier-level techniques are orthogonal to our approach and can be combined with MR$^2$ in a modular way. To demonstrate this, we incorporate LA into MR$^2$ and evaluate the hybrid method on the standard long-tail benchmark CIFAR-100-IB-100 (Cao et al., 2019) using ResNet-32. The results in Table 7 show that incorporating MR$^2$ with the imbalanced-learning method LA further improves overall accuracy and reduces performance disparity.

Table 7: Applying our MR$^2$ to CIFAR-100-IB-100.

| Model | Overall | Easy | Medium | Hard |
|---|---|---|---|---|
| ERM | 37.7 | 69.6 | 38.9 | 5.4 |
| LA | 41.9 | 66.9 | 43.9 | 15.6 |
| LA + MR$^2$ | 44.2 | 68.1 | 46.3 | 18.9 |

# B MISSING PROOFS

## B.1 CONNECTION BETWEEN $\ell_{\bar{s}}$ AND $\hat{\ell}_{\bar{s}}$

In this section, we prove that

$$\ell_{\bar{s}}(f, \mathbf{x}, y) = \ln\left[1 + \sum_{\mathbf{x}^+ \in \mathcal{D}_y \setminus \{\mathbf{x}\}} \exp(\|\phi(\mathbf{x}) - \phi(\mathbf{x}^+)\|_2^2 - 2\bar{s})\right] \tag{18}$$

is a differentiable relaxation of

$$\hat{\ell}_{\bar{s}}(f, \mathbf{x}, y) = \max\left[0, \max_{\mathbf{x}^+ \in \mathcal{D}_y \setminus \{\mathbf{x}\}} \left(\|\phi(\mathbf{x}) - \phi(\mathbf{x}^+)\|_2^2 - 2\bar{s}\right)\right]. \tag{19}$$

Let $z_j = \|\phi(\mathbf{x}) - \phi(\mathbf{x}_j^+)\|_2^2 - 2\bar{s}$, where $j \in [m]$ and $m = |\mathcal{D}_y \setminus \{\mathbf{x}\}|$. Denote $z_{m+1} = 0$. We can rewrite $\ell_{\bar{s}}$ and $\hat{\ell}_{\bar{s}}$ as:

$$\ell_{\bar{s}}(f, \mathbf{x}, y) = \ln\left[\sum_{j=1}^{m+1} \exp(z_j)\right], \quad \hat{\ell}_{\bar{s}}(f, \mathbf{x}, y) = \max_{j \in [m+1]} z_j \tag{20}$$

According to the property of LogSumExp, we have:

$$\hat{\ell}_{\bar{s}}(f, \mathbf{x}, y) = \max_{j \in [m+1]} z_j = \ln \exp(\max_{j \in [m+1]} z_j) \leq \ln\left[\sum_{j=1}^{m+1} \exp(z_j)\right] = \ell_{\bar{s}}(f, \mathbf{x}, y). \tag{21}$$

Therefore, $\ell_{\bar{s}}$ is a smooth approximation to the $\hat{\ell}_{\bar{s}}$. $\qquad\square$

## B.2 CONNECTION BETWEEN PAIRWISE DISTANCE AND MEAN SQUARED DEVIATION

For random vectors $(\mathbf{x}, \mathbf{y})$ independently drawn from some distribution, let $\boldsymbol{\mu} = \mathbb{E}[\mathbf{x}]$ denote the mean and define the expected squared deviation as $\|\mathbf{s}\|_2^2 = \mathbb{E}[\|(\mathbf{x} - \boldsymbol{\mu})\|_2^2]$. Then

$$\mathbb{E}[\|\mathbf{x} - \mathbf{y}\|_2^2] = \mathbb{E}[\|(\mathbf{x} - \boldsymbol{\mu}) - (\mathbf{y} - \boldsymbol{\mu})\|_2^2] \tag{22}$$

$$= \mathbb{E}[\|(\mathbf{x} - \boldsymbol{\mu})\|_2^2] + \mathbb{E}[\|(\mathbf{y} - \boldsymbol{\mu})\|_2^2] - 2\mathbb{E}[(\mathbf{x} - \boldsymbol{\mu})^\top (\mathbf{y} - \boldsymbol{\mu})] \tag{23}$$

$$= 2\mathbb{E}[\|(\mathbf{x} - \boldsymbol{\mu})\|_2^2] = 2\|\mathbf{s}\|_2^2. \tag{24}$$

First equality in Eq. (24) follows from the independence of $(\mathbf{x} - \boldsymbol{\mu})$ and $(\mathbf{y} - \boldsymbol{\mu})$.

## B.3 PROOF OF LEMMA 1

**Restated Lemma (Lemma 1: $\gamma$-margin bound).** *Let $\mathcal{F}$ be a set of real valued functions. Given $\gamma > 0$, then for any $\delta > 0$, with probability at least $1 - \delta$, for all $f \in \mathcal{F}$:*

$$\mathcal{R}(f) \leq \widehat{\mathcal{R}}_{\mathcal{D}}^{\gamma}(f) + 4\sqrt{2K}\widehat{\mathfrak{R}}_{\mathcal{D}}^{\gamma}(\mathcal{F}) + 3\sqrt{\frac{\ln\frac{2}{\delta}}{2N}} \tag{25}$$

*Proof.* Let $\mathcal{G}$ be a set of real valued functions that map labeled instances $(\mathbf{x}, y)$ into the $\gamma$-margin loss:

$$\mathcal{G} = \{(\mathbf{x}, y) \mapsto \ell_{\gamma}(f, \mathbf{x}, y) \mid f \in \mathcal{F}\} \tag{26}$$

Since $\mathcal{G}$ maps into $[0, 1]$, applying Theorem 3.3 from Mohri et al. (2018) yields that, with probability at least $1 - \delta$, the following holds for all $g \in \mathcal{G}$:

$$\mathbb{E}_{\mathbf{x}, y}[g(\mathbf{x}, y)] \leq \frac{1}{N}\sum_{i=1}^{N} g(\mathbf{x}_i, y_i) + 2\widehat{\mathfrak{R}}_{\mathcal{D}}(\mathcal{G}) + 3\sqrt{\frac{\ln\frac{2}{\delta}}{2N}} \tag{27}$$

By substituting the definition of $g$, we obtain

$$\mathbb{E}_{\mathbf{x},y}[\ell_\gamma(f,\mathbf{x},y)] \le \frac{1}{N}\sum_{i=1}^N \ell_\gamma(f,\mathbf{x}_i,y_i) + 2\widehat{\mathfrak{R}}_\mathcal{D}(\mathcal{G}) + 3\sqrt{\frac{\ln\frac{2}{\delta}}{2N}} \tag{28}$$

Recall the definitions of the empirical and expected $\gamma$-margin risks: $\widehat{\mathcal{R}}_\mathcal{D}^\gamma(f) = \frac{1}{N}\sum_{i=1}^N \ell_\gamma(f,\mathbf{x}_i,y_i)$ and $\mathcal{R}^\gamma(f) = \mathbb{E}_{\mathbf{x},y}[\ell_\gamma(f,\mathbf{x},y)]$. Using the fact that the 0-1 loss is upper-bounded by $\gamma$-margin loss, we have $\mathcal{R}(f) \le \mathcal{R}^\gamma(f)$. Thus, applying Eq. (28), we obtain

$$\mathcal{R}(f) \le \widehat{\mathcal{R}}_\mathcal{D}^\gamma(f) + 2\widehat{\mathfrak{R}}_\mathcal{D}(\mathcal{G}) + 3\sqrt{\frac{\ln\frac{2}{\delta}}{2N}} \tag{29}$$

Next, we aim to bound $\widehat{\mathfrak{R}}_\mathcal{D}(\mathcal{G})$ using $\widehat{\mathfrak{R}}_\mathcal{D}(\mathcal{F})$. $\widehat{\mathfrak{R}}_\mathcal{D}(\mathcal{G})$ can be written as:

$$\widehat{\mathfrak{R}}_\mathcal{D}(\mathcal{G}) = \frac{1}{N}\mathbb{E}_{\boldsymbol{\sigma}}\left[\sup_{f\in\mathcal{F}}\sum_{i=1}^N \sigma_i\ell_\gamma(f,\mathbf{x}_i,y_i)\right] \tag{30}$$

$$= \frac{1}{N}\mathbb{E}_{\boldsymbol{\sigma}}\left[\sup_{f\in\mathcal{F}}\sum_{i=1}^N \sigma_i\Phi_{\gamma_{y_i}}(\gamma_f(\mathbf{x}_i,y_i))\right] \tag{31}$$

$$\le \frac{1}{N}\mathbb{E}_{\boldsymbol{\sigma}}\left[\sup_{f\in\mathcal{F}}\sum_{i=1}^N \sigma_i\frac{\gamma_f(\mathbf{x}_i,y_i)}{\gamma_{y_i}}\right], \tag{32}$$

where $\boldsymbol{\sigma} = (\sigma_i)_i$ with $\sigma_i$ being i.i.d Rademacher variables sampled uniformly from $\{-1,1\}$. Eq.(32) holds from the $\frac{1}{\gamma}$-Lipschitz continuity of $\Phi_\gamma$. We define the function $\Psi_i$ as:

$$\Psi_i(f) = \frac{\gamma_f(\mathbf{x}_i,y_i)}{\gamma_{y_i}} = \frac{f(\mathbf{x}_i,y_i) - \max_{y'\neq y_i}f(\mathbf{x}_i,y')}{\gamma_{y_i}}. \tag{33}$$

Then, for any $f, f' \in \mathcal{F}$, we have

$$|\Psi_i(f) - \Psi_i(f')| = \frac{1}{\gamma_{y_i}}|(f(\mathbf{x}_i,y_i) - \max_{y'\neq y_i}f(\mathbf{x}_i,y')) - (f'(\mathbf{x}_i,y_i) - \max_{y'\neq y_i}f'(\mathbf{x}_i,y'))| \tag{34}$$

$$\le \frac{1}{\gamma_{y_i}}|\max_{y'\neq y_i}(f(\mathbf{x}_i,y_i) - f'(\mathbf{x}_i,y_i)) + (f'(\mathbf{x}_i,y') - f(\mathbf{x}_i,y'))| \tag{35}$$

$$\le \frac{2}{\gamma_{y_i}}\max_{y\in\mathcal{Y}}|f(\mathbf{x}_i,y) - f'(\mathbf{x}_i,y)| \tag{36}$$

$$\le \frac{2}{\gamma_{y_i}}\sum_{y\in\mathcal{Y}}|f(\mathbf{x}_i,y) - f'(\mathbf{x}_i,y)| \tag{37}$$

$$\le \frac{2\sqrt{K}}{\gamma_{y_i}}\sqrt{\sum_{y\in\mathcal{Y}}|f(\mathbf{x}_i,y) - f'(\mathbf{x}_i,y)|^2} \tag{38}$$

Let $a_{y'} = f'(\mathbf{x}_i,y')$, $b_{y'} = f(\mathbf{x}_i,y')$, and $y^* = \arg\max_{y'\neq y} a_{y'}$, Eq.(35) holds from the sub-additivity of the maximum:

$$\max_{y'\neq y_i}f'(\mathbf{x},y') - \max_{y'\neq y_i}f(\mathbf{x}_i,y') = a_{y^*} - \max_{y'\neq y_i}f(\mathbf{x}_i,y') \le a_{y^*} - b_{y^*} \le \max_{y'\neq y_i}(a_{y'} - b_{y'}). \tag{39}$$

We have Eq.(38) because of the $L_1$-$L_2$ norm inequality for $K$-dimensional $\mathbf{x}$ vector: $\|\mathbf{x}\|_1 \le \sqrt{K}\|\mathbf{x}\|_2$. From Eq.(38), it is evident that $\Psi_i$ is $\frac{2\sqrt{K}}{\gamma_{y_i}}$-Lischitz with respect to the $\|\cdot\|_2$ norm. Thus, by the vector contraction lemma (Lemma 5 of Cortes et al. (2016)), we have:

$$\widehat{\mathfrak{R}}_\mathcal{D}(\mathcal{G}) \le \frac{1}{N}\mathbb{E}_{\boldsymbol{\sigma}}\left[\sup_{f\in\mathcal{F}}\sum_{i=1}^N \sigma_i\Psi_i(f)\right] \le \frac{2\sqrt{2K}}{N}\mathbb{E}_{\boldsymbol{\epsilon}}\left[\sup_{f\in\mathcal{F}}\left\{\sum_{k=1}^K\sum_{i\in I_k}\sum_{y\in\mathcal{Y}}\epsilon_{ij}\frac{f(\mathbf{x}_i,y)}{\gamma_k}\right\}\right] = 2\sqrt{2K}\widehat{\mathfrak{R}}_\mathcal{D}^\gamma(\mathcal{F}) \tag{40}$$

Combining Eq.(40) and Eq.(29), we arrive at Eq.(25). $\qquad\square$

### B.4 Proof of Lemma 2

**Restated Lemma (Lemma 2: Bound on the empirical Rademacher complexity).** *For non-negative $\gamma > 0$, consider $\mathcal{F} = \{(\mathbf{x}, y) \mapsto \mathbf{w}_y^\top \phi(\mathbf{x}) \mid \mathbf{w}_y \in \mathbb{R}^d, \|\mathbf{w}_y\|_2 \leq \Lambda\}$. Let $\{\|\hat{\boldsymbol{\mu}}\|_2^2\}_{k=1}^K$ and $\{\|\hat{\mathbf{s}}\|_2^2\}_{k=1}^K$ denote the per-class squared norms of the feature means and the mean squared deviations, respectively, as defined in Section 3. Then, the following bound holds for all $f \in \mathcal{F}$:*

$$\widehat{\mathfrak{R}}_{\mathcal{D}}^\gamma(\mathcal{F}) \leq \Lambda \sqrt{\frac{K}{N}} \sqrt{\sum_{k=1}^K \frac{\|\hat{\boldsymbol{\mu}}_k\|_2^2 + \|\hat{\mathbf{s}}_k\|_2^2}{\gamma_k^2}} \tag{41}$$

*Proof.*

$$\widehat{\mathfrak{R}}_{\mathcal{D}}^\gamma(\mathcal{F}) = \frac{1}{N} \mathbb{E}_{\boldsymbol{\epsilon}} \left[ \sup_{f \in \mathcal{F}} \left\{ \sum_{k=1}^K \sum_{i \in I_k} \sum_{y \in \mathcal{Y}} \epsilon_{iy} \frac{f(\mathbf{x}_i, y)}{\gamma_k} \right\} \right] \tag{42}$$

$$= \frac{1}{N} \mathbb{E}_{\boldsymbol{\epsilon}} \left[ \sup_{\|\mathbf{w}_y\|_2 \leq \Lambda} \left\{ \sum_{k=1}^K \sum_{i \in I_k} \sum_{y \in \mathcal{Y}} \epsilon_{iy} \frac{\mathbf{w}_y^\top \phi(\mathbf{x}_i)}{\gamma_k} \right\} \right] \tag{43}$$

$$= \frac{1}{N} \mathbb{E}_{\boldsymbol{\epsilon}} \left[ \sup_{\|\mathbf{w}_y\|_2 \leq \Lambda} \left\{ \sum_{y \in \mathcal{Y}} \mathbf{w}_y^\top \sum_{k=1}^K \sum_{i \in I_k} \epsilon_{iy} \frac{\phi(\mathbf{x}_i)}{\gamma_k} \right\} \right] \tag{44}$$

$$\leq \frac{1}{N} \mathbb{E}_{\boldsymbol{\epsilon}} \left[ \sup_{\|\mathbf{w}_y\|_2 \leq \Lambda} \left\{ \sum_{y \in \mathcal{Y}} \|\mathbf{w}_y\|_2 \left\| \sum_{k=1}^K \sum_{i \in I_k} \epsilon_{iy} \frac{\phi(\mathbf{x}_i)}{\gamma_k} \right\|_2 \right\} \right] \tag{45}$$

$$= \frac{\Lambda}{N} \mathbb{E}_{\boldsymbol{\epsilon}} \left[ \sum_{y \in \mathcal{Y}} \left\| \sum_{k=1}^K \sum_{i \in I_k} \epsilon_{iy} \frac{\phi(\mathbf{x}_i)}{\gamma_k} \right\|_2 \right] \tag{46}$$

$$= \frac{\Lambda}{N} \sum_{y \in \mathcal{Y}} \mathbb{E}_{\boldsymbol{\epsilon}} \left[ \left\| \sum_{k=1}^K \sum_{i \in I_k} \epsilon_{iy} \frac{\phi(\mathbf{x}_i)}{\gamma_k} \right\|_2 \right] \tag{47}$$

$$\leq \frac{\Lambda}{N} \sum_{y \in \mathcal{Y}} \sqrt{\mathbb{E}_{\boldsymbol{\epsilon}} \left[ \left\| \sum_{k=1}^K \sum_{i \in I_k} \epsilon_{iy} \frac{\phi(\mathbf{x}_i)}{\gamma_k} \right\|_2^2 \right]} \tag{48}$$

$$= \frac{\Lambda}{N} \sum_{y \in \mathcal{Y}} \sqrt{\mathbb{E}_{\boldsymbol{\epsilon}} \left[ \sum_{k,i} \sum_{k',i'} \epsilon_{iy} \epsilon_{i'y} \frac{\phi(\mathbf{x}_i)^\top \phi(\mathbf{x}_{i'})}{\gamma_k \gamma_{k'}} \right]} \tag{49}$$

$$= \frac{K\Lambda}{N} \sqrt{\sum_{k=1}^K \frac{\sum_{i \in I_k} \|\phi(\mathbf{x}_i)\|_2^2}{\gamma_k^2}} \tag{50}$$

$$= \Lambda \sqrt{\frac{K}{N}} \sqrt{\sum_{k=1}^K \frac{\|\hat{\boldsymbol{\mu}}_k\|_2^2 + \|\hat{\mathbf{s}}_k\|_2^2}{\gamma_k^2}} \tag{51}$$

Eq.(45) holds by Cauchy-Schwarz inequality. We have Eq.(48) by Jensen's inequality, *i.e.*, $\mathbb{E}[\|X\|_2] \leq \sqrt{\mathbb{E}[\|X\|_2^2]}$. Eq.(49) follows using the fact that $\mathbb{E}[\epsilon_i \epsilon_{i'}] = 0$ for $i \neq i'$ and $\mathbb{E}[\epsilon_i \epsilon_i] = 1$. Given the

class-balanced setting, *i.e.*, $|I_k| = \frac{N}{K}$, Eq.(51) holds by the following equality:

$$\sum_{i \in I_k} \|\phi(\mathbf{x}_i)\|_2^2 = \sum_{i \in I_k} ((\phi(\mathbf{x}_i) - \hat{\boldsymbol{\mu}}_k) + \hat{\boldsymbol{\mu}}_k)^\top ((\phi(\mathbf{x}_i) - \hat{\boldsymbol{\mu}}_k) + \hat{\boldsymbol{\mu}}_k) \tag{52}$$

$$= \sum_{i \in I_k} [\|\phi(\mathbf{x}_i) - \hat{\boldsymbol{\mu}}_k\|_2^2] + \sum_{i \in I_k} [\|\hat{\boldsymbol{\mu}}_k\|_2^2] + 2 \underbrace{\sum_{i \in I_k} [(\phi(\mathbf{x}_i) - \hat{\boldsymbol{\mu}}_k)^\top \hat{\boldsymbol{\mu}}_k]}_{=0} \tag{53}$$

$$= |I_k| \cdot \frac{1}{|I_k|} \sum_{i \in I_k} [\|\phi(\mathbf{x}_i) - \hat{\boldsymbol{\mu}}_k\|_2^2] + |I_k| \cdot \|\hat{\boldsymbol{\mu}}_k\|_2^2 \tag{54}$$

$$= \frac{N}{K} (\|\hat{\mathbf{s}}_k\|_2^2 + \|\hat{\boldsymbol{\mu}}_k\|_2^2) \tag{55}$$

$\square$

## B.5 PROOF OF LEMMA 3

**Lemma 4 (Log–sum–exp dominates the ramp).** *For every $\gamma > 0$ and all $u \in \mathbb{R}$,*

$$\Phi_\gamma(u) = \min(1, \max(0, 1 - \frac{u}{\gamma})) \leq \log_2\big(1 + \exp\{-\frac{u}{\gamma}\}\big).$$

*Proof.* Let $g(u)$ denote the RHS: $g(u) = \log_2(1 + \exp\{-\frac{u}{\gamma}\})$. It is strictly decreasing and convex because $g'(u) = -\frac{1}{\gamma \ln 2} \frac{\exp\{-u/\gamma\}}{1+\exp\{-u/\gamma\}} < 0$ and $g''(u) = \frac{1}{\gamma^2 \ln 2} \frac{\exp\{-y/\gamma\}}{(1+\exp\{-u/\gamma\})^2} > 0$. We verify the inequality in three disjoint regions:

- $u \geq \gamma$. $\Phi_\gamma(u) = 0$ by definition, while $g(u) > 0$. Hence the inequality holds.

- $u \leq 0$. Now $\Phi_\gamma(u) = 1$. Since $g(u)$ is decreasing, $g(u) \geq g(0) = \log_2 2 = 1$.

- $0 < u < \gamma$. In this interval, let $t = \frac{u}{\gamma} \in (0, 1)$. We aim to prove that

$$h(t) = \log_2(1 + \exp\{-t\}) - (1 - t) = g(u) - \Phi_\gamma(u) \geq 0, \ \forall t \in (0, 1). \tag{56}$$

Let $\sigma(t) = \frac{1}{1+e^{-t}}$ be the sigmoid function, the derivative of $h(t)$ is:

$$h'(t) = \frac{1}{\ln 2} \frac{-e^{-t}}{1 + e^{-t}} + 1 = (1 - \frac{1}{\ln 2}) + \frac{\sigma(t)}{\ln 2} \tag{57}$$

Since $\sigma(t)$ is monotonically increasing, $h'(t) > (1 - \frac{1}{\ln 2}) + \frac{\sigma(0)}{\ln 2} = 1 - \frac{1}{2 \ln 2} \approx 0.279 > 0$. Therefore, $h(t)$ is strictly increasing for $t \in (0, 1)$. Since $h(0) = \log_2(2) - 1 = 0$, we have $h(t) > 0, \ \forall t \in (0, 1)$

Combining the three regions proves the claim. $\square$

**Restated Lemma (Lemma 3).** *For non-negative $\gamma > 0$. Let $\widehat{\mathcal{R}}_{\mathcal{D}}^\gamma(f) = \frac{1}{N} \sum_{\mathbf{x}, y \in \mathcal{D}} [\ell_\gamma(f, \mathbf{x}, y)]$ and $\widehat{\mathcal{R}}_{\mathcal{D}}^{\gamma,\text{ce}}(f) = \frac{1}{N} \sum_{\mathbf{x}, y \in \mathcal{D}} [\ell_{\gamma,\text{ce}}(f, \mathbf{x}, y)]$. Then, the following bound holds for all $f \in \mathcal{F}$:*

$$\widehat{\mathcal{R}}_{\mathcal{D}}^\gamma(f) \leq \frac{1}{\ln 2} \widehat{\mathcal{R}}_{\mathcal{D}}^{\gamma,\text{ce}}(f) \tag{58}$$

*Proof.* Let $\mathbf{z} = [f(\mathbf{x}, 1), ..., f(\mathbf{x}, K)]^\top$ denote the logit vector. Using our Lemma 4 that $\Phi_\gamma(u) \leq \log_2(1 + \exp\{-\frac{u}{\gamma}\})$, and the fact that $\log_2(a) = \ln(a)/\ln(2)$

$$\ell_\gamma(f, \mathbf{x}, y) = \Phi_{\gamma_y}(\gamma_f(\mathbf{x}, y)) \leq \log_2\left[1 + \exp(-\frac{\gamma_f(\mathbf{x}, y)}{\gamma_y})\right] = \frac{1}{\ln 2} \ln\left[1 + \exp(-\frac{\gamma_f(\mathbf{x}, y)}{\gamma_y})\right] \tag{59}$$

Because $\max_j a_j \leq \ln \sum_j \exp(a_j)$, we have

$$\ln \exp(-\frac{\gamma_f(\mathbf{x}, y)}{\gamma_y}) = -\frac{\gamma_f(\mathbf{x}, y)}{\gamma_y} = \max_{k \neq y}[\frac{\mathbf{z}_k - \mathbf{z}_y}{\gamma_y}] \leq \ln \left[\sum_{k \neq y} \exp(-\frac{\mathbf{z}_y - \mathbf{z}_k}{\gamma_y})\right] \tag{60}$$

Adding 1 inside the logarithm on both sides yields

$$\ell_{\boldsymbol{\gamma}}(f, \mathbf{x}, y) \leq \frac{1}{\ln 2} \ln \left[1 + \exp(-\frac{\gamma_f(\mathbf{x}, y)}{\gamma_y})\right] \tag{61}$$

$$\leq \frac{1}{\ln 2} \ln \left[1 + \sum_{k \neq y} \exp(-\frac{\mathbf{z}_y - \mathbf{z}_k}{\gamma_y})\right] \tag{62}$$

$$= \frac{1}{\ln 2} \ell_{\boldsymbol{\gamma},\text{ce}}(f, \mathbf{x}, y) \tag{63}$$

Since $\ell_{\boldsymbol{\gamma}}$ is point-wise upper-bounded by $\frac{1}{\ln 2} \ell_{\boldsymbol{\gamma},\text{ce}}$, it follows immediately that

$$\widehat{\mathcal{R}}_{\mathcal{D}}^{\boldsymbol{\gamma}}(f) \leq \frac{1}{\ln 2} \widehat{\mathcal{R}}_{\mathcal{D}}^{\boldsymbol{\gamma},\text{ce}}(f). \tag{64}$$

$\square$

### B.6 Proof of Corollary 1

**Restated Corollary (Corollary 1 Optimal choice of $\boldsymbol{\gamma}$).** *For fixed $\bar{c}$, the complexity term of the bound is minimized when*

$$\gamma_y = \frac{\bar{c}K(\|\hat{\boldsymbol{\mu}}_y\|_2^2 + \|\hat{\mathbf{s}}_y\|_2^2)^{\frac{1}{3}}}{\sum_{k=1}^{K}(\|\hat{\boldsymbol{\mu}}_k\|_2^2 + \|\hat{\mathbf{s}}_k\|_2^2)^{\frac{1}{3}}} \quad \text{for all } y \in [K]. \tag{65}$$

*Proof.* To minimize the complexity term in Eq. (13) under the total margin constraint $c$, we solve the following constrained optimization problem:

$$\boldsymbol{\gamma} = \underset{\boldsymbol{\gamma}=[\gamma_k]}{\arg\min} \sum_{k=1}^{K} \frac{\|\hat{\boldsymbol{\mu}}_k\|_2^2 + \|\hat{\mathbf{s}}_k\|_2^2}{\gamma_k^2} \quad \text{s.t.} \sum_{k \in [K]} \gamma_k = c. \tag{66}$$

Denote $\alpha_k = \|\hat{\boldsymbol{\mu}}_k\|_2^2 + \|\hat{\mathbf{s}}_k\|_2^2 > 0$. It is equivalent to solve the following Lagrangian problem with multiplier $\upsilon \in \mathbb{R}$.

$$\min_{\boldsymbol{\gamma}} \max_{\upsilon} \sum_{k=1}^{K} \frac{\alpha_k}{\gamma_k^2} + \upsilon(\sum_{k \in [K]} \gamma_k - c) \tag{67}$$

Setting the partial derivatives w.r.t. $\gamma_y$ to zero gives

$$-\frac{2\alpha_y}{\gamma_y^3} + \upsilon = 0 \implies \gamma_y = (\frac{2\alpha_y}{\upsilon})^{\frac{1}{3}}. \tag{68}$$

Thus every optimal $\gamma_y$ is proportional to $\alpha_y^{1/3}$. Since $\frac{1}{K} \sum_k \gamma_k = \bar{c}$, we have

$$\bar{c}K = (2\frac{\sum_k \alpha_k}{\upsilon})^{\frac{1}{3}} \implies \upsilon^{\frac{1}{3}} = \frac{(2 \sum_k \alpha_k)^{\frac{1}{3}}}{\bar{c}K} \tag{69}$$

It follows directly that:

$$\gamma_y = \frac{\bar{c}K\alpha_y^{\frac{1}{3}}}{\sum_k \alpha_k^{\frac{1}{3}}} \tag{70}$$

$\square$

### B.7 PROOF OF PROPOSITION 2

We begin with lemmas leading to the proposition of our general bound.

**Lemma 5.** *If $X_1, ..., X_n$ are independent sub-Gaussian r.v.s with variance proxies $\sigma_1^2, ..., \sigma_n^2$, then $Z = \sum_{i=1}^n X_i$ is sub-Gaussian with variance proxy $\sum_{i=1}^n \sigma_i^2$.*

**Lemma 6.** *(**Maximal Inequality** Rinaldo (2018)) Suppose $X_1, ..., X_n$ are sub-Gaussian r.v.s with variance proxy $\sigma^2$. Then*

$$\mathbb{E}[\max_{i \in [n]} X_i] \leq \sqrt{2\sigma^2 \log(n)} \tag{71}$$

**Restated Proposition (Proposition 2: General bound on the $\gamma$-margin Rademacher complexity).**
*Fix conjugate exponents $p, q \in [1, \infty]$ such that $1/p + 1/q = 1$. Let $\mathcal{F}_q = \{(x, y) \mapsto \mathbf{w}_y^\top \phi(\mathbf{x}) \mid \mathbf{w}_y \in \mathbb{R}^d, \|\mathbf{w}_y\|_q \leq \Lambda_q\}$. For each class $k \in [K]$, we write the average $L_p^2$ norm as*

$$r_{k,p}^2 = \frac{1}{|I_k|} \sum_{i \in I_k} \|\phi(\mathbf{x}_i)\|_p^2. \tag{72}$$

*Then, for all $f \in \mathcal{F}_q$, the empirical $\gamma$-margin Rademacher complexity satisfies*

$$\widehat{\mathfrak{R}}_\mathcal{D}^\gamma(\mathcal{F}_q) \leq \begin{cases} \sqrt{2\log(d)}\Lambda_q \sqrt{\frac{K}{N}} \sqrt{\sum_{k=1}^K \frac{r_{k,\infty}^2}{\gamma_k^2}} & \text{if } p = \infty, \\ 2^{\frac{p}{2}} \frac{\Gamma(\frac{p+1}{2})}{\sqrt{\pi}} \Lambda_q \sqrt{\frac{K}{N}} \sqrt{\sum_{k=1}^K \frac{r_{k,p}^2}{\gamma_k^2}} & \text{if } 2 < p < \infty, \\ \Lambda_q \sqrt{\frac{K}{N}} \sqrt{\sum_{k=1}^K \frac{r_{k,p}^2}{\gamma_k^2}} & \text{if } 1 \leq p \leq 2, \end{cases} \tag{73}$$

*where $\Gamma$ is the gamma function.*

*Proof.* Using Hölder's inequality and the bound on $\|\mathbf{w}\|_p$, we have

$$\widehat{\mathfrak{R}}_\mathcal{D}^\gamma(\mathcal{F}_q) = \frac{1}{N} \mathbb{E}_{\boldsymbol{\epsilon}} \left[ \sup_{f \in \mathcal{F}_q} \left\{ \sum_{k=1}^K \sum_{i \in I_k} \sum_{y \in \mathcal{Y}} \epsilon_{iy} \frac{f(\mathbf{x}_i, y)}{\gamma_k} \right\} \right] \tag{74}$$

$$= \frac{1}{N} \mathbb{E}_{\boldsymbol{\epsilon}} \left[ \sup_{\|\mathbf{w}_y\|_q \leq \Lambda} \left\{ \sum_{y \in \mathcal{Y}} \mathbf{w}_y^\top \sum_{k=1}^K \sum_{i \in I_k} \epsilon_{iy} \frac{\phi(\mathbf{x}_i)}{\gamma_k} \right\} \right] \tag{75}$$

$$\leq \frac{1}{N} \mathbb{E}_{\boldsymbol{\epsilon}} \left[ \sup_{\|\mathbf{w}_y\|_q \leq \Lambda} \left\{ \sum_{y \in \mathcal{Y}} \|\mathbf{w}_y\|_q \left\| \sum_{k=1}^K \sum_{i \in I_k} \epsilon_{iy} \frac{\phi(\mathbf{x}_i)}{\gamma_k} \right\|_p \right\} \right] \tag{76}$$

$$= \frac{K\Lambda}{N} \mathbb{E}_{\boldsymbol{\epsilon}} \left[ \left\| \sum_{k=1}^K \sum_{i \in I_k} \epsilon_i \frac{\phi(\mathbf{x}_i)}{\gamma_k} \right\|_p \right] \tag{77}$$

We use the shorthand $\mathbf{u}$ as

$$\mathbf{u} = \sum_{k=1}^K \sum_{i \in I_k} \epsilon_i \frac{\phi(\mathbf{x}_i)}{\gamma_k} \in \mathbb{R}^d, \tag{78}$$

and $u_j$ as the $j$-th coordinate of $\mathbf{u}$:

$$u_j = \sum_{k=1}^K \sum_{i \in I_k} \epsilon_i \frac{\phi(\mathbf{x}_i)_j}{\gamma_k}, \tag{79}$$

Using Lemma 5 and the fact that a Rademacher variable is 1-sub-Gaussian, the proxy variance $\sigma_j^2$ of $u_j$ is

$$\sigma_j^2 = \sum_{k=1}^K \frac{\sum_{i \in I_k} \phi(\mathbf{x}_i)_j^2}{\gamma_k^2} \leq \sum_{k=1}^K \frac{\sum_{i \in I_k} \|\phi(\mathbf{x}_i)\|_p^2}{\gamma_k^2} = \frac{N}{K} \sum_{k=1}^K \frac{r_{k,p}^2}{\gamma_k^2} \tag{80}$$

The inequality holds for any $p \geq 1$, as for any vector $\mathbf{v} \in \mathbb{R}^d$ and $j \in [d]$, we have $|\mathbf{v}_j| \leq \|\mathbf{v}\|_p$.

**Proof of upper bound, case $p = \infty$.**

$$\widehat{\mathfrak{R}}_{\mathcal{D}}^{\gamma}(\mathcal{F}_1) \leq \frac{K\Lambda}{N} \mathbb{E}_{\boldsymbol{\epsilon}}[\|\mathbf{u}\|_{\infty}] \tag{81}$$

$$= \frac{K\Lambda}{N} \mathbb{E}_{\boldsymbol{\epsilon}} \left[ \max_{j \in [d], s \in \{-1, +1\}} s u_j \right] \tag{82}$$

Thus, by Maximal Inequality Lemma (Lemma 6),

$$\widehat{\mathfrak{R}}_{\mathcal{D}}^{\gamma}(\mathcal{F}_1) \leq \sqrt{2 \log(d)} \Lambda \sqrt{\frac{K}{N}} \sqrt{\sum_{k=1}^{K} \frac{r_{k,\infty}^2}{\gamma_k^2}} \tag{83}$$

**Proof of upper bound, case $1 \leq p < \infty$.**

By Khintchine's inequality Haagerup (1981), the following holds:

$$\mathbb{E}_{\boldsymbol{\epsilon}}[\|\mathbf{u}\|_p] = \left[ \sum_{j \in [d]} \mathbb{E}|u_j|^p \right]^{\frac{1}{p}} \leq C_p (\sum_{j \in [d]} \sigma_j^2)^{\frac{1}{2}} = C_p \sqrt{\frac{N}{K} \sum_{k=1}^{K} \frac{r_{k,p}^2}{\gamma_k^2}} \tag{84}$$

Therefore, we have:

$$\widehat{\mathfrak{R}}_{\mathcal{D}}^{\gamma}(\mathcal{F}_q) \leq C_p \Lambda \sqrt{\frac{K}{N}} \sqrt{\sum_{k=1}^{K} \frac{r_{k,p}^2}{\gamma_k^2}}, \tag{85}$$

where $C_p = 1$ for $p \leq 2$, and

$$C_p = 2^{\frac{p}{2}} \frac{\Gamma(\frac{p+1}{2})}{\sqrt{\pi}} \tag{86}$$

for $p > 2$. $\qquad\qquad\square$

### B.8 PER-CLASS ERROR BOUND

We bound the per-class error $\mathcal{R}_k$ for class $k$ w.p. at least $1 - \delta$:

$$\mathcal{R}_k(f) \leq \frac{1}{\ln 2} \widehat{\mathcal{R}}_{\mathcal{D}_k}^{\gamma, \mathbf{ce}} + \frac{4}{\gamma_k} \widehat{\mathfrak{R}}_k(\mathcal{F}) + \varepsilon(\delta, N), \tag{87}$$

where $\widehat{\mathfrak{R}}_k \propto \sqrt{\|\hat{\boldsymbol{\mu}}_k\|_2^2 + \|\hat{\mathbf{s}}_k\|_2^2}$ and $\epsilon$ is a low-order term. It suffices to express $\widehat{\mathfrak{R}}_k(\mathcal{F})$ as the empirical Rademacher complexity restricted to class $k$:

$$\widehat{\mathfrak{R}}_k(\mathcal{F}) = \frac{1}{|I_k|} \mathbb{E}_{\boldsymbol{\epsilon}} \left[ \sup_{f \in \mathcal{F}} \sum_{i \in I_k} \epsilon_i f(\mathbf{x}_i, y_i) \right], \tag{88}$$

where $\boldsymbol{\epsilon}$ are Rademacher variables.

*Proof.* By definition,

$$\widehat{\mathfrak{R}}_k(\mathcal{F}_k) = \frac{1}{N_k} \mathbb{E}_{\boldsymbol{\epsilon}} \left[ \sup_{\|\mathbf{w}\|_2 \leq \Lambda} \sum_{i \in I_k} \epsilon_i \mathbf{w}^{\top} \phi(\mathbf{x}_i) \right] \tag{89}$$

$$= \frac{1}{N_k} \mathbb{E}_{\boldsymbol{\epsilon}} \left[ \sup_{\|\mathbf{w}\|_2 \leq \Lambda} \mathbf{w}^{\top} \sum_{i \in I_k} \epsilon_i \phi(\mathbf{x}_i) \right]. \tag{90}$$

Applying Cauchy–Schwarz,

$$\widehat{\mathfrak{R}}_k(\mathcal{F}_k) \leq \frac{\Lambda}{N_k} \mathbb{E}_{\boldsymbol{\epsilon}} \left[ \left\| \sum_{i \in I_k} \epsilon_i \phi(\mathbf{x}_i) \right\|_2 \right]. \tag{91}$$

Now note that each coordinate of $u = \sum_{i \in I_k} \epsilon_i \phi(\mathbf{x}_i)$ is a Rademacher sum with variance proxy $\sum_{i \in I_k} \phi(\mathbf{x}_i)_j^2$. Thus,

$$\mathbb{E}_{\boldsymbol{\epsilon}} \| u \|_2^2 = \sum_{j=1}^d \mathbb{E} \left[ \left( \sum_{i \in I_k} \epsilon_i \phi(\mathbf{x}_i)_j \right)^2 \right] \tag{92}$$

$$= \sum_{j=1}^d \sum_{i \in I_k} \phi(\mathbf{x}_i)_j^2 \tag{93}$$

$$= \sum_{i \in I_k} \| \phi(\mathbf{x}_i) \|_2^2. \tag{94}$$

Taking square roots,

$$\widehat{\mathfrak{R}}_k(\mathcal{F}_k) \leq \frac{\Lambda}{N_k} \sqrt{\sum_{i \in I_k} \| \phi(\mathbf{x}_i) \|_2^2}. \tag{95}$$

Finally, decomposing into squared mean and variance gives

$$\frac{1}{N_k} \sum_{i \in I_k} \| \phi(\mathbf{x}_i) \|_2^2 = \| \hat{\boldsymbol{\mu}}_k \|_2^2 + \| \hat{\mathbf{s}}_k \|_2^2.$$

Therefore,

$$\widehat{\mathfrak{R}}_k(\mathcal{F}) \leq \Lambda \sqrt{\| \hat{\boldsymbol{\mu}}_k \|_2^2 + \| \hat{\mathbf{s}}_k \|_2^2}.$$

$\square$

## C  ADDITIONAL EXPERIMENTAL DETAILS AND RESULTS

### C.1  ADDITIONAL DETAILS OF BASELINE METHODS

We adopt the official implementations of the baselines to ensure fair comparison, with the exception of CSR Jin et al. (2025), DRO Sagawa et al. (2020), and FairDRO Jung et al. (2023), where differences in settings require modifications, as described below. CSR is designed to enhance robust fairness under adversarial attacks. To align it with our clean-data training setting, we remove the adversarial perturbations during training. DRO and FairDRO target worst-group performance. In our setting without group labels, we substitute group labels with class labels.

### C.2  ADDITIONAL IMPLEMENTATION DETAILS

**Implementation details for CIFAR-100.** Following Cao et al. (2019), we use simple data augmentation: each image is padded by 4 pixels on all sides, then a $32 \times 32$ crop is randomly sampled from the padded image or its horizontal flip. The model is trained using stochastic gradient descent with momentum 0.9, weight decay $5 \times 10^{-4}$, a batch size of 128, and for 300 epochs. The initial learning rate is set to 0.1 and follows a cosine decay schedule. Experiments are performed using a single NVIDIA A100-40GB GPU.

**Implementation details for training from scratch on ImageNet.** We train the ResNet-50 in 300 epochs with a cosine and batch size of 128 on 8 NVIDIA A100-40GB GPUs. The initial learning rate is set to 0.1 and the weight decay is set to $10^{-4}$. We use the SGD optimizer with a momentum of 0.9.

**Implementation details for CLIP ResNet-50 and CLIP ViT-B/32.** Following the fine-tuning configurations of WiSE-FT Wortsman et al. (2022), we use the default PyTorch AdamW optimizer for

Table 8: Effect of different $L_p$ norms on CIFAR-100 with ResNet-32. We observe consistent improvements across all metrics for different $p$ values.

| $p$ | 1 | 3 | 5 | 7 | 9 |
|---|---|---|---|---|---|
| Overall | 73.4 (+2.6) | 73.7 (+2.9) | 73.9 (+3.1) | 73.7 (+2.9) | 73.2 (+2.4) |
| Easy | 85.8 (+1.3) | 85.9 (+1.4) | 85.9 (+1.4) | 85.9 (+1.4) | 85.8 (+1.3) |
| Medium | 73.5 (+2.5) | 73.7 (+2.7) | 73.8 (+2.8) | 73.6 (+2.6) | 73.4 (+2.4) |
| Hard | 60.7 (+4.0) | 61.4 (+4.7) | 61.9 (+5.2) | 61.5 (+4.8) | 60.2 (+3.5) |

Table 9: Effect of different norms on ImageNet with CLIP ResNet-50. For the cosine classifier models, we note consistent improvement across all metrics for different $p$ values.

| $p$ | 1 | 3 | 5 | 7 | 9 |
|---|---|---|---|---|---|
| Overall | 76.5 (+0.9) | 77.1 (+1.5) | 76.9 (+1.3) | 77.1 (+1.5) | 76.8 (+1.2) |
| Easy | 91.6 (+0.2) | 91.7 (+0.3) | 91.5 (+0.1) | 91.7 (+0.3) | 91.5 (+0.1) |
| Medium | 79.3 (+0.9) | 79.9 (+1.5) | 79.6 (+1.2) | 79.9 (+1.5) | 79.4 (+1.0) |
| Hard | 58.8 (+1.8) | 59.8 (+2.8) | 60.0 (+3.0) | 59.7 (+2.7) | 59.6 (+2.6) |

10 epochs with weight decay of 0.1. We set an initial learning rate of $3 \times 10^{-5}$ and follows a cosine decay schedule with 500 warm-up steps. We use a batch size of 512 and minimal data augmentation as in Radford et al. (2021); Wortsman et al. (2022). Since CLIP models are cosine classifier models, we set $p = 3$ for $L_p$ norm. Experiments are performed using four NVIDIA A100-40GB GPUs.

**Implementation details for MAE ViT-B/16.** Following the fine-tuning setup of He et al. (2022), we use the default AdamW optimizer with base learning rate of $10^{-3}$ for 100 epochs. The batch size is set to 1024 and cosine decay learning schedule with 5 warmup epochs. For fair comparison, we avoid advanced augmentation strategies (*e.g.*, RandAugment, CutMix and AutoAug) and use only standard augmentations such as random crop and flip. Experiments are performed using eight NVIDIA A100-40GB GPUs.

**Implementation details for MoCov2 ResNet-50.** Using a pretrained MoCov2 He et al. (2020) backbone, we train a supervised linear classifier on frozen features. The classifier is optimized with SGD using a learning rate of 30, momentum of 0.9, and a batch size of 256. Training runs for 100 epochs, with the learning rate decayed by a factor of 0.1 at epochs 60 and 80. We apply standard data augmentations, including random cropping and horizontal flipping. Experiments are performed using a single NVIDIA A100-40GB GPU.

### C.3 ADDITIONAL EXPERIMENTAL RESULTS

**Effect of different norms on CIFAR-100 and ImageNet.** Table 8 and Table 9 report the results with various $L_p$ norms on CIFAR-100 and ImageNet. As expected, using $p \geq 1$ consistently improves both balanced accuracies and generalization, supporting our theoretical framework.

**Details numerical results for StanfordCars, OxfordPets, Flowers, Food and FGVCAircraft.** In Table 10, we present all the metrics for five fine-grained datasets. We observe significant improvement on "hard" classes and moderate gains on "easy" classes, indicating the effectiveness of our $MR^2$ in improving balanced accuracies and generalization.

**Additional main component analysis.** In Table 11, we present additional analysis on the effects of the logit margin loss $\ell_{\gamma,ce}$ and the representation margin loss $\ell_{\bar{s}}$. The results show that both regularization components are effective in enhancing balanced accuracies and overall performance.

**Numerical results for hyper-parameters $\bar{c}$ and $\lambda$.** The detailed values are in Table 12 and Table 13.

**Per-class accuracies of our $MR^2$.** As datasets such as CIFAR-100 and ImageNet involve a large number of classes, making tabulated reporting less clear, we use CIFAR-10 for clarity. Additionally, we train our models on 10% of the CIFAR-10 training set to ensure meaningful class-wise variation, as training on the full CIFAR-10 dataset leads to near-saturated performance. The results in Table 14 clearly indicate that our $MR^2$ method improves performance most significantly for the harder-to-

Table 10: Fairness improvements on StanfordCars, OxfordPets, Flowers, Food and FGVCAircraft.

|  | Metric | Cars | Pets | Flowers | Food | Aircraft |
|---|---|---|---|---|---|---|
| CLIP RN-50 | Overall | 75.8 | 89.2 | 96.7 | 81.4 | 34.5 |
|  | Easy | 92.0 | 97.4 | 100 | 91.0 | 65.7 |
|  | Medium | 77.9 | 90.6 | 99.3 | 82.9 | 29.4 |
|  | Hard | 57.4 | 79.7 | 90.9 | 70.2 | 8.4 |
| + $\mathrm{MR}^2$ | Overall | 76.4 (+0.6) | 90.4 (+1.2) | 97.3 (+0.6) | 81.9 (+0.5) | 35.5 (+1.0) |
|  | Easy | 92.2 (+0.2) | 97.8 (+0.4) | 100 (+0.0) | 91.1 (+0.1) | 65.8 (+0.1) |
|  | Medium | 78.4 (+0.5) | 91.7 (+1.1) | 99.9 (+0.6) | 83.3 (+0.4) | 30.0 (+0.6) |
|  | Hard | 58.7 (+1.3) | 81.6 (+1.9) | 92.2 (+1.3) | 71.3 (+1.1) | 10.8 (+2.4) |

Table 11: Additional main component analysis. "RN", "P", "C" and "M" stand for ResNet, PreAct, CLIP and MAE, respectively.

(a) **CIFAR-100**

|  | Model | Overall | Easy | Medium | Hard |
|---|---|---|---|---|---|
| RN-20 | $\ell_{ce}$ | 68.7 | 83.3 | 70.3 | 53.0 |
|  | $\ell_{\gamma,ce}$ | 70.3 | 84.1 (+0.8) | 71.7 (+1.4) | 55.8 (+2.8) |
|  | $\ell_{\gamma,ce} + \lambda\ell_{\bar{s}}$ | 70.9 | 84.4 (+1.1) | 72.2 (+1.9) | 56.7 (+3.7) |
| PRN-20 | $\ell_{ce}$ | 69.1 | 83.8 | 70.5 | 53.4 |
|  | $\ell_{\gamma,ce}$ | 70.2 | 84.2 (+0.4) | 71.8 (+1.3) | 55.1 (+1.7) |
|  | $\ell_{\gamma,ce} + \lambda\ell_{\bar{s}}$ | 71.3 | 84.7 (+0.9) | 73.2 (+2.7) | 56.6 (+3.2) |
| PRN-32 | $\ell_{ce}$ | 71.2 | 85.0 | 71.7 | 56.7 |
|  | $\ell_{\gamma,ce}$ | 72.7 | 85.8 (+0.8) | 73.4 (+1.7) | 59.2 (+2.5) |
|  | $\ell_{\gamma,ce} + \lambda\ell_{\bar{s}}$ | 73.3 | 86.1 (+1.1) | 73.9 (+2.2) | 60.1 (+3.4) |

(b) **ImageNet**

|  | Model | Overall | Easy | Medium | Hard |
|---|---|---|---|---|---|
| CRN-50 | $\ell_{ce}$ | 75.2 | 91.1 | 78.3 | 56.4 |
|  | $\ell_{\gamma,ce}$ | 75.9 | 91.2 (+0.1) | 78.6 (+0.3) | 58.1 (+1.7) |
|  | $\ell_{\gamma,ce} + \lambda\ell_{\bar{s}}$ | 76.9 | 91.5 (+0.4) | 79.7 (+1.4) | 59.6 (+3.2) |
| CViT-B | $\ell_{ce}$ | 75.6 | 91.4 | 78.4 | 57.0 |
|  | $\ell_{\gamma,ce}$ | 76.3 | 91.5 (+0.1) | 78.7 (+0.3) | 58.9 (+1.9) |
|  | $\ell_{\gamma,ce} + \lambda\ell_{\bar{s}}$ | 77.1 | 91.7 (+0.3) | 79.9 (+1.5) | 59.8 (+2.8) |
| MViT-B | $\ell_{ce}$ | 80.4 | 94.5 | 83.9 | 62.7 |
|  | $\ell_{\gamma,ce}$ | 81.3 | 94.6 (+0.1) | 84.5 (+0.6) | 65.0 (+2.3) |
|  | $\ell_{\gamma,ce} + \lambda\ell_{\bar{s}}$ | 82.0 | 94.8 (+0.3) | 85.3 (+1.4) | 66.1 (+3.4) |

classify classes. For example, the two worst-performing classes, bird and cat, see improvements of 8.2% and 9.1% in accuracy, respectively. This aligns with the goal of reducing class-wise performance disparity.

**Effect of $\bar{c}$ on ImageNet.** In Figure 8, we study the impact of $\bar{c}$ using CLIP ResNet-50 on ImageNet. Both curves exhibit similar trends, with optima located around $\bar{c} = 2$.

**Effect of $\lambda$ on ImageNet.** In Figure 9, we examine the effect of $\lambda$ using CLIP ResNet-50 on ImageNet. $\mathrm{MR}^2$ achieves optimal overall and hard class performance around $\lambda = 0.4$, and the performance remains stable in this region.

**Standard deviations.** In Table 15, we report the standard deviations across diverse model architectures on CIFAR-10 and ImageNet.

**Visualization of the features.** Figure 10 visualizes the effect of our loss on the feature geometry of hard CIFAR-100 classes. We randomly selected hard classes and extract their last-layer test features. These features are then projected to 2D using t-SNE van der Maaten & Hinton (2008). Under standard cross-entropy (Figure 10a), the clusters of different classes are highly entangled and many hard classes exhibit large within-class spread. In contrast, with our $\mathrm{MR}^2$ loss (Figure 10b), the features of each hard class become much more compact and different classes are better separated.

# D SOCIETAL IMPACT AND LIMITATION

Disparities in class-wise accuracy, even under class-balanced training, pose fairness risks in real-world deployments. Our work contributes toward building more trustworthy and fair machine learning systems. Although $\mathrm{MR}^2$ focuses on class-level disparity, it does not account for unfairness arising from data-related issues such as label noise and class co-occurrence.

Table 12: Numerical values for Figure 5.

| $\bar{c}$ | 0 | 0.5 | 1 | 2 | 3 | 5 |
|---|---|---|---|---|---|---|
| Hard | 60.4 | 60.8 | 61.6 | 61.9 | 61.2 | 61.0 |
| Overall | 72.8 | 73.2 | 73.7 | 73.9 | 73.4 | 73.1 |

Table 13: Numerical values for Figure 6.

| $\lambda$ | 0.01 | 0.05 | 0.1 | 0.3 | 0.5 | 0.7 | 0.9 |
|---|---|---|---|---|---|---|---|
| Hard | 60.7 | 61.0 | 61.7 | 61.8 | 61.9 | 61.6 | 61.5 |
| Overall | 73.1 | 73.5 | 73.9 | 73.9 | 73.9 | 73.8 | 73.7 |

Table 14: Per-class accuracies on CIFAR-10. $\Delta$ denotes the relative improvements.

| | automobile | truck | ship | frog | horse | airplane | deer | dog | bird | cat |
|---|---|---|---|---|---|---|---|---|---|---|
| ERM | 91.4 | 87.1 | 85.5 | 80.4 | 79.0 | 76.5 | 76.1 | 70.5 | 70.1 | 62.5 |
| $\mathrm{MR}^2$ (ours) | 92.1 | 89.0 | 87.4 | 83.0 | 82.5 | 78.9 | 79.2 | 77.3 | 78.3 | 71.6 |
| $\Delta$ | +0.7 | +1.9 | +1.9 | +2.6 | +3.5 | +2.4 | +3.1 | +6.8 | +8.2 | +9.1 |

# E    LLM USAGE STATEMENT

We used ChatGPT only for minor language editing to improve clarity and conciseness. No part of the research idea, methodology, or analysis was generated by LLMs.

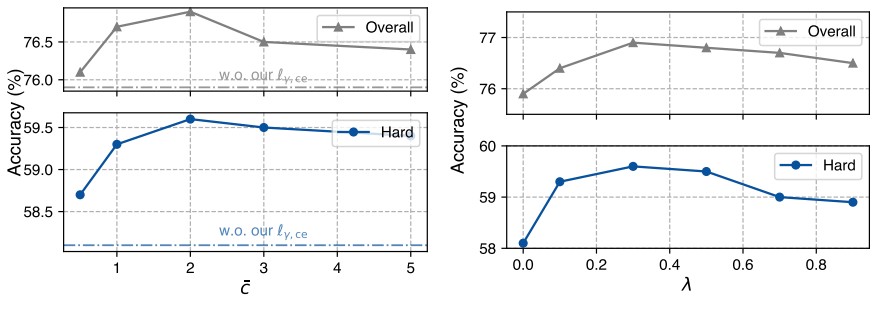

Figure 8: Effect of $\bar{c}$.          Figure 9: Effect of $\lambda$.

Table 15: Standard deviations across diverse model architectures on CIFAR-10 and ImageNet.

(a) **CIFAR-100**

| Model | Hard | Medium | Easy | Overall |
|---|---|---|---|---|
| ResNet-20 | 0.0033 | 0.0029 | 0.0019 | 0.0035 |
| + $\mathrm{MR}^2$ | 0.0012 | 0.0027 | 0.0016 | 0.0022 |
| PreAct RN-20 | 0.0016 | 0.0030 | 0.0033 | 0.0019 |
| + $\mathrm{MR}^2$ | 0.0027 | 0.0039 | 0.0015 | 0.0024 |
| ResNet-32 | 0.0029 | 0.0036 | 0.0019 | 0.0032 |
| + $\mathrm{MR}^2$ | 0.0019 | 0.0029 | 0.0037 | 0.0017 |
| PreAct RN-32 | 0.0026 | 0.0016 | 0.0034 | 0.0039 |
| + $\mathrm{MR}^2$ | 0.0030 | 0.0012 | 0.0037 | 0.0022 |
| W-RN-22-10 | 0.0025 | 0.0038 | 0.0019 | 0.0034 |
| + $\mathrm{MR}^2$ | 0.0013 | 0.0027 | 0.0040 | 0.0022 |

(b) **ImageNet**

| Model | Hard | Medium | Easy | Overall |
|---|---|---|---|---|
| ResNet-50 TfS | 0.0023 | 0.0039 | 0.0015 | 0.0028 |
| + $\mathrm{MR}^2$ | 0.0018 | 0.0029 | 0.0032 | 0.0015 |
| MoCov2 RN-50 | 0.0015 | 0.0013 | 0.0024 | 0.0008 |
| + $\mathrm{MR}^2$ ($\ell_{\gamma,\mathrm{ce}}$) | 0.0019 | 0.0032 | 0.0011 | 0.0027 |
| CLIP RN-50 | 0.0036 | 0.0018 | 0.0039 | 0.0015 |
| + $\mathrm{MR}^2$ | 0.0029 | 0.0035 | 0.0012 | 0.0023 |
| CLIP ViT-B/32 | 0.0019 | 0.0017 | 0.0020 | 0.0028 |
| + $\mathrm{MR}^2$ | 0.0021 | 0.0033 | 0.0029 | 0.0037 |
| MAE ViT-B/16 | 0.0018 | 0.0022 | 0.0034 | 0.0017 |
| + $\mathrm{MR}^2$ | 0.0026 | 0.0038 | 0.0023 | 0.0035 |

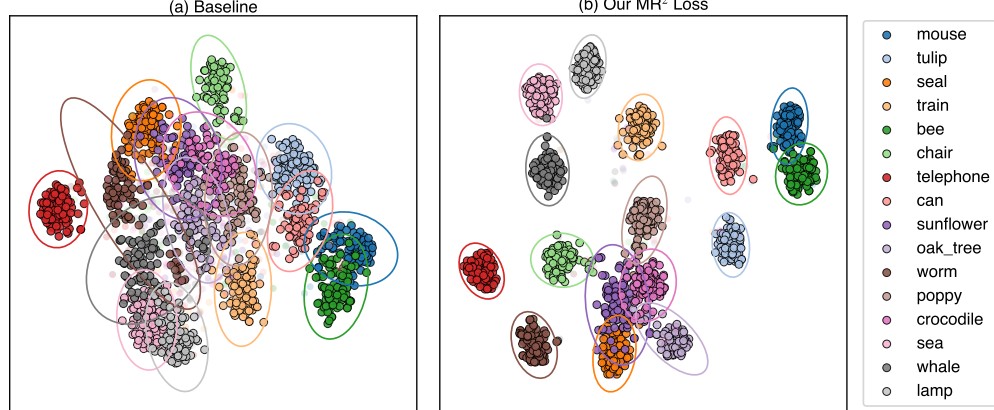

Figure 10: t-SNE visualizations of feature representations for randomly selected hard CIFAR-100 classes using (a) standard cross-entropy and (b) our $\mathrm{MR}^2$.

