# OpenReview forum: "Reducing Class-Wise Performance Disparity via Margin Regularization"
_ICLR.cc/2026/Conference — ICLR 2026 Poster_

### Official Review · Reviewer_dj7P · 2025-10-28

**Soundness:** 3
**Presentation:** 2
**Contribution:** 3
**Rating:** 6
**Confidence:** 3

**Summary:**

The paper presents MR$^2$, a novel objective function for training DNNs with reduced inter-class performance disparity and improved generalization. The MR$^2$ objective consists of a margin scaled cross-enropy loss and representation regularization term for encouraging intra-class feature compactness. It's shown that the proposed methods is connected to logit margin loss and contrastive learning and can tighted generalization bound. Extensive experimental results on image classification are provided to validate the effectiveness of the proposed method.

**Strengths:**

1. The paper addresses a largely overlooked issue: inter-class performance disparity under class-balanced setting.
2. The paper provideds extensive theoretical analysis to justify the proposed method by linking it to $\gamma$-margin loss and generalization bound.
3. Comprehensive experimental results are presented.

**Weaknesses:**

1. The loss in Eq. 1 can be regarded as cross-entropy loss with class-dependent temperature scaling. Temperature scaling has been studied a lot in knowledge distillation and model calibration. It's better to discuss and compare with some of those methods such as [1].
2. In experiments, it's better to compare with more methods that promote class separability, such as [2], [3] and [4].
3. The statement "hard classes ... to such classes" in Line 137~140 seems to assume that $||\hat{\mu}_y||$ is roughly constant across all classes so the margin mainly depends on $||\hat{s}_y||$. It's better to provide evidence to support that $||\hat{\mu}_y||$ roughly stays constant across classes.
4. All experiments on ImageNet are based on fine-tuning. It's better to provide some results for training from scratch on ImageNet, even just for small models such as ResNet-18 and ViT-tiny.
5. It's mentioned that advanced augmentation strategies are avoided in experiments. However, it's better to show results with those augmentations to show the effectiveness of the proposed method when combined with modern augmentaion settings.

[1] K. Xu et al. Feature Normalized Knowledge Distillation for Image Classification. ECCV 2020.
[2] Y. Wen et al. A discriminative feature learning approach for deep face recognition. ECCV 2016.
[3] W. Wan et al. Rethinking Feature Distribution for Loss Functions in Image Classification. CVPR 2018.
[4] EH. Yang et al. Conditional mutual information constrained deep learning for classification. TNNLS 2025.

**Questions:**

1. Would it be better to make $\gamma$ trainable after initialization according to Eq. 1?
2. In Fig. 5, how can $\bar{c}$ be 0? This will lead to a "dividing by 0" error in Eq. 1.

---

> ### Author Response · Authors · 2025-11-21
> **Response to Reviewer dj7P**
>
> > W1. The loss in Eq. 1 can be regarded as cross-entropy loss with class-dependent temperature scaling. Temperature scaling has been studied a lot in knowledge distillation and model calibration. It's better to discuss and compare with some of those methods such as [1].
>
> **Reply**:  We thank the reviewer for this point.
> In the revised version (Line 214-230, Sec.3.1), we explicitly discuss temperature scaling methods and clarify how our class-dependent temperature differs from standard calibration/distillation approaches.
>
> “Our logit-level loss in Eq. (5) can also be viewed as introducing a *class-dependent temperature*, similar to temperature scaling used in knowledge distillation and model calibration (Xu et al., 2020; Guo et al., 2017), where a (typically global) temperature is applied to smooth or calibrate prediction confidences(calibration aims to make predicted confidences better match true accuracies). By contrast, our $\boldsymbol{\gamma}$ is *spread-aware* and is explicitly designed to balance margins between easy and hard classes, rather than improving calibration quality.”
>
> > W2. In experiments, it's better to compare with more methods that promote class separability, such as [2], [3] and [4].
>
> **Reply**: We thank the reviewer for this suggestion. In the revised version, we have included [2–4] as additional baselines and compared them against our method in the main results (Table 1). We observe that these baselines are less effective than our method in promoting balanced class-wise performance.
>
> > W3. The statement "hard classes ... to such classes" in Line 137~140 seems to assume that $\|\mu_y\|$ is roughly constant across all classes so the margin mainly depends on $\|s_y\|$. It's better to provide evidence to support that $\|\mu_y\|$  roughly stays constant across classes.
>
> **Reply**: We thank the reviewer for this insightful comment. In the revised version (Sec.A, Fig. 8), we plot the per-class values of $||\hat{\boldsymbol{\mu}}_y||_2$ and $||\hat{\mathbf{s}}_y||_2$ on ImageNet. The results show that $||\hat{\boldsymbol{\mu}}_y||_2$  exhibits much smaller variation across classes compared to $||\hat{\mathbf{s}}_y||_2$, supporting our assumption that the margin differences are primarily driven by the spread term $||\hat{\mathbf{s}}_y||_2$ rather than by large fluctuations in $||\hat{\boldsymbol{\mu}}_y||_2$.
>
> > W4. All experiments on ImageNet are based on fine-tuning. It's better to provide some results for training from scratch on ImageNet, even just for small models such as ResNet-18 and ViT-tiny.
>
> To further address the reviewer’s concern, we additionally train ResNet-50 *from scratch* on ImageNet for 300 epochs using our loss. The results are shown in the following table.
>
> | Model        | Overall | Easy                | Medium              | Hard                |
> |-------------|---------|---------------------|---------------------|---------------------|
> | RN-50 train from scratch   | 71.7    | 88.5                | 74.1                | 52.6                |
> | + MR$^2$    | 74.2    | 89.9 **(+1.4)**     | 76.9 **(+2.8)**     | 55.9 **(+3.3)**     |
>
>
> We have also included the results and training details in the revised version (Table 2).
> These results confirm that our method remains effective even when training from scratch on ImageNet.

---

> ### Author Response · Authors · 2025-11-21
>
> > W5. It's mentioned that advanced augmentation strategies are avoided in experiments. However, it's better to show results with those augmentations to show the effectiveness of the proposed method when combined with modern augmentation settings.
>
> We thank the reviewer for this suggestion.
> To address this concern, we additionally apply advanced data augmentation strategies, including RandAugment and AutoAugment.
> The corresponding results are reported below and have also been added to Sec.A Table 5 in the revised version.
>
> **Table**  MR$^2$ with advanced data augmentations on ImageNet using CLIP-ResNet-50.
>
> | Model                          | Overall              | Easy                    | Medium                  | Hard                    |
> |--------------------------------|----------------------|-------------------------|-------------------------|-------------------------|
> | Standard data augmentation     | 75.2                 | 91.1                    | 78.3                    | 56.4                    |
> | + MR$^2$                        | 76.9 **(+1.7)**      | 91.5 **(+0.4)**         | 79.7 **(+1.4)**         | 59.6 **(+3.2)**         |
> | RandAug                        | 75.8                 | 91.5                    | 78.8                    | 57.2                    |
> | + MR$^2$                        | 77.3 **(+1.5)**      | 92.0 **(+0.5)**         | 79.8 **(+1.0)**         | 60.1 **(+2.9)**         |
> | AutoAug                        | 76.2                 | 91.9                    | 79.2                    | 57.5                    |
> | + MR$^2$                        | 78.1 **(+1.9)**      | 92.3 **(+0.4)**         | 81.0 **(+1.8)**         | 61.1 **(+3.5)**         |
>
> Again, MR$^2$ consistently boosts the accuracy of ``hard’’ classes while providing mild improvements for easy ones, thereby reducing performance disparity without introducing trade-offs. These results confirm that our loss continues to perform well when combined with modern data augmentation techniques.
>
> > Q1. Would it be better to make $\gamma$  trainable after initialization according to Eq. 1?
>
> **Reply**: We do not find this beneficial. Since our loss $\ell_{\boldsymbol{\gamma},\mathsf{ce}}$ is monotonically increasing in $\boldsymbol{\gamma}$, treating $\boldsymbol{\gamma}$ as a trainable parameter would introduce a trivial descent direction: the optimizer can always reduce the loss by shrinking $\boldsymbol{\gamma}$.
> As a result, $\boldsymbol{\gamma}$ would systematically converge toward small values, collapsing the class-dependent margins and failing to enforce a balanced margin across easy and hard classes.  We have added this explanation in the revised version (Lines 758-774)
>
> > Q2. In Fig. 5, how can $\bar{c}$ be 0? This will lead to a "dividing by 0" error in Eq. 1.
>
> In Fig. 5, the case $\bar{c}=0$ is used only as a reference baseline and corresponds to *not using our logit-level loss*.
> We do not plug $\bar{c}=0$ into Eq.(1); Eq.(1) is applied only for $\bar{c}>0$.
> We apologize for the confusion and have clarified this point in the revised version.

---

> > ### Comment · Reviewer_dj7P · 2025-11-26
> >
> > Thank the authors for their detailed response, which has addressed my major concerns. Thus, I will keep my positive score.

---

> > > ### Author Response · Authors · 2025-11-27
> > >
> > > Dear Reviewer dj7P
> > >
> > > Thank you for your valuable feedback and for taking the time to review our updated work. We greatly appreciate your recognition—it means a lot to us and motivates us to keep improving.
> > >
> > > Your valuable comments helped us refine the connection to temperature scaling, clarify the fluctuations of $\mu_y$, and strengthen the experiments (additional baselines, training-from-scratch results, and advanced data augmentations).
> > >
> > > Thank you again for your constructive comments and encouragement. We sincerely appreciate your support.
> > >
> > > Sincerely,
> > > Authors

---

### Official Review · Reviewer_5Lxp · 2025-11-01

**Soundness:** 4
**Presentation:** 4
**Contribution:** 3
**Rating:** 6
**Confidence:** 4

**Summary:**

This study aims to address the problem of class disparity that arises in classification under balanced settings. To this end, the authors propose regularizers that control the margins in both the logit and representation spaces, assigning larger margins to more challenging classes to achieve balance in the error bound. Furthermore, the paper provides theoretical evidence supporting the effectiveness of the proposed approach through well-formulated theorems and corresponding proofs. In addition, extensive empirical evaluations across various datasets, including fine-grained datasets, as well as architectures such as CLIP, demonstrate the effectiveness of the proposed method

**Strengths:**

* By providing background on the class disparity problem—an increasingly critical issue in modern classification settings—and analyzing its underlying causes, this study effectively establishes the motivation for addressing this problem

* The study also supports the validity of the proposed approach with solid theoretical analysis, comprising precisely stated theorems and corresponding proofs

* Furthermore, extensive experiments conducted on a wide range of datasets, including fine-grained benchmarks, and on architectures such as CLIP, confirm the robustness of the proposed approach. Ablation studies (e.g., main component analysis, effect of $\bar{c}$ and $\lambda$) also demonstrate its effectiveness

**Weaknesses:**

**W1.** As illustrated in Eq. 13 (lines 286–291), there exists a trade-off between the first and second terms depending on the value of $\gamma$. Although this trade-off is bounded through Corollary 1, further tuning of the coefficient $\bar{c}$ is still required. This remains a tuning issue, in combination with another hyperparameter $\lambda$, which increases the overall burden of hyperparameter tuning.

**W2.** The proposed approach indirectly verifies its effectiveness in addressing the class disparity problem through performance improvement. However, it would be more convincing to more directly validate whether the margins for hard classes have indeed increased, using metrics such as the class margin introduced in [R1].

**W3.** The related work section provides a thorough review of not only other works on class disparity but also long-tail learning, which is a similar area of work in that it focuses on controlling class margins. However, despite the presence of conceptually related approaches in hyperspherical learning—such as those that separately control the margins between classes and samples or adjust margins according to sample difficulty, as shown in [R1,R2]—the paper does not include a survey or discussion of hyperspherical learning.


[R1] Zhou et al., Learning Towards the Largest Margins, ICLR 2022

[R2] Son et al., Difficulty-aware Balancing Margin Loss for Long-tailed Recognition, AAAI 2025

**Questions:**

**Q1** (w.r.t **W1**). Ablation studies on CIFAR-100 using ResNet-32 were conducted to examine hyperparameter tuning; however, since they are limited to a single dataset, providing additional insights could help alleviate concerns about tuning sensitivity. For instance, is there any guideline on how to adjust hyperparameter values—such as increasing or decreasing them—depending on the number of samples or labels?

**Q2.** Although the proposed method shows improved performance compared to other approaches across various experiments, the class disparity still appears to remain severe. Could the authors provide a more detailed explanation of why the issue has not been completely resolved, along with potential limitations or directions for future work to address it?

**Q3.** How many times were the experiments conducted with different random seeds? To address concerns about potential cherry-picking and demonstrate the robustness of the results across different random environments, please provide this information.


**Things to improve the paper that did not impact the score**:

* (lines 022-023) typo: "demonstrate demonstrate"

* (line 800) In Eq. 39, $a'\_{y}$ and $b'\_{y}$ newly appear without being mentioned or defined (it seems a typo error for $a\_{y'}$ and $b\_{y'}$)

---

> ### Author Response · Authors · 2025-11-21
> **Response to Reviewer 5Lxp**
>
> > W1. As illustrated in Eq. 13 (lines 286–291), there exists a trade-off between the first and second terms depending on the value of $\gamma$. Although this trade-off is bounded through Corollary 1, further tuning of the coefficient $\bar{c}$ is still required. This remains a tuning issue, in combination with another hyperparameter $\lambda$, which increases the overall burden of hyperparameter tuning.
>
> **Reply:** We thank the reviewer for raising this concern.
>
> In practice, we observe that for any positive choices of $\bar{c}$ and $\lambda$, **incorporating our two loss terms consistently improves over ERM, and hyperparameter tuning only adjusts how large this improvement is.**
> In addition, we provide (1) a default value for $\bar{c}$ with an intuitive justification, and (2) empirical evidence that the $\lambda$ curve is not sensitive.
>
> First, a safe and effective default is to set $\bar{c}=1$, which makes the average margin scale comparable to that of the standard cross-entropy loss.
> Figure 5 shows that both overall accuracy and hard-class accuracy remain stable over a reasonably wide range of $\bar{c}$ around this value (roughly $\bar{c}\in[1,3]$), and, importantly, *all* these settings consistently outperform the ERM baseline.
> Thus, only very coarse tuning is needed, and even suboptimal choices of $\bar{c}$ still yield positive gains.
>
> Second, combining $\bar{c}$ with the coefficient $\lambda$ does not substantially increase the tuning burden.
> As shown in Figure 6, performance is quite flat over a broad range of $\lambda$ values, again with our method consistently improving over ERM, indicating that our approach is robust to this choice as well.
>
> Furthermore, as shown in our response to Q1 (deferred below), the behavior of $\lambda$ and $\bar{c}$ on other datasets such as ImageNet is very similar to that on CIFAR-100 (Fig. 9 and Fig. 10), further demonstrating that our method is consistently insensitive to these hyperparameters across datasets and that hyperparameter tuning is not critical to obtaining improvements.
>
> > W2. The proposed approach indirectly verifies its effectiveness in addressing the class disparity problem through performance improvement. However, it would be more convincing to more directly validate whether the margins for hard classes have indeed increased, using metrics such as the class margin introduced in [R1].
>
> **Reply**: We thank the reviewer for raising this point. In Table 4 of the revised version (or the table below), we also study the effect of our MR$^2$ on output-level and classifier-level margins (formal definition of the two margins is in the main paper). As in Table 4, MR$^2$ significantly enlarges output-level margins across all subsets, reduces the margin gap between easy and hard class, and raises the classifier-level margin from 58.4 to 81.8.
>
> | Method | Output-level margin Avg. ↑ | Output-level margin Easy ↑ | Output-level margin Med. ↑ | Output-level margin Hard ↑ | Classifier-level margin ↑ |
> |--------|----------------|---------------|------|----|------------|
> | ERM    | 0.158          | 0.236         | 0.144         | 0.093          | 58.4       |
> | MRR    | 0.373          | 0.435         | 0.396         | 0.289          | 81.8       |
>
> > W3.  Hyperspherical learning and related margin-based methods.
>
> **Reply**: We thank the reviewer for pointing out the connection to hyperspherical learning works such as Zhou et al. [R1] and Son et al. [R2].  We have explicitly discussed them in the revised version (Lines 100-102).
>
> Zhou et al. ([R1]) and Son et al. ([R2]) are related in that they also operate in a hyperspherical / margin-based regime, but their motivations, methods, and theoretical focus are quite different from ours. Zhou et al.  propose a principled framework for maximizing global class and sample margins on the hypersphere and unify existing margin-softmax losses via rigorous “largest margin’’ analysis. Their theory primarily does not explicitly address class-wise performance disparity or class-dependent spread. In contrast, DBM is an algorithmic design for long-tailed recognition that combines a class-wise margin based on label frequency with an instance-wise margin for hard positives; it is motivated by practical long-tail performance and provides geometric intuition but no generalization bounds or margin-theoretic guarantees. Our work is complementary: instead of maximizing a single global margin or focusing on long-tail class frequency, we directly model and control the class-wise feature spread and margin disparity between easy and hard classes, and we derive class-conditional margin bounds that guide the design of our regularizer.

---

> ### Author Response · Authors · 2025-11-21
>
> > Q1. Ablation studies on CIFAR-100 using ResNet-32 were conducted to examine hyperparameter tuning; however, since they are limited to a single dataset, providing additional insights could help alleviate concerns about tuning sensitivity. For instance, is there any guideline on how to adjust hyperparameter values—such as increasing or decreasing them—depending on the number of samples or labels?
>
> **Reply:** To address this concern, we additionally conduct the same hyperparameter ablation studies on ImageNet; the results are reported in Fig. 9 and Fig. 10 in the revised version.
> We observe that the optimal values of $\bar{c}$ and $\lambda$ on ImageNet are similar to those on CIFAR-100.
> We suggest a simple guideline: tuning $\bar{c}$ around 2 and $\lambda$ around 0.3 is a good empirical choice, as the performance in these regions is strong and relatively insensitive to small changes across datasets.
>
> > Q2: Could the authors provide a more detailed explanation of why the issue has not been completely resolved, along with potential limitations or directions for future work to address it?
>
> **Reply**: We thank the reviewer for this question. As discussed in our limitations (Sec.~C), our algorithm cannot  resolve disparity arising from *noisy supervision* and*spurious correlation*  related issues, such as label noise ((incorrect or ambiguous labels)) and class co-occurrence (images where multiple object classes or strongly correlated context appear simultaneously). In addition, classes at different levels of semantic granularity are inherently different in recognition difficulty; for example, distinguishing a generic “dog’’ class is much easier than recognizing a specific breed such as “chihuahua’’.
>
> Our method targets the portion of class-wise disparity that is attributable to representation geometry and margin imbalance, but it cannot correct discrepancies that are fundamentally induced by noisy or ambiguous supervision. A promising future direction is to combine our approach with improved data curation or complementary training frameworks (e.g., active learning that selectively acquire labels for ambiguous samples to help clean up fuzzy class boundary).
>
> We have included the discussion in the revised version (Lines 896-940).
>
> > Q3: How many times were the experiments conducted with different random seeds?
>
> **Reply**: All experiments are repeated three times. *We adhere to the standard code of conduct for submissions and ensure that no cherry-picking is involved.* The detailed standard deviations across diverse architectures, pre-trained models have been added to Table 15 in the revised version.
>
> > Typos.
>
> **Reply**: We thank the reviewer for pointing these out; both typos have been fixed.

---

> > ### Comment · Reviewer_5Lxp · 2025-11-27
> >
> > ### Acknowledgement to the Authors
> >
> > ---
> >
> > I appreciate the clarifications regarding **W1**, **W3**, and **Q2**, as well as the additional efforts related to **W2**, **Q1**, and **Q3**. The revisions were instrumental in addressing the primary concerns about the explicit validation and versatility of the proposed method.
> >
> > ### Responses to the Rebuttal
> >
> > ---
> >
> > **W1 & Q1.**
> > Thank you for providing clear guidelines to address the concerns about hyperparameter tuning and for extending the experiments to ImageNet to demonstrate the scalability of the proposed approach. I am pleased that these additional experiments further support the generalizability of the suggested guidelines.
> >
> > **W2.**
> > Thank you for conducting the explicit quantitative evaluation of the margin as requested. This metric helps substantiate both the effectiveness of the proposed method and the authors’ claims more objectively. Additionally, it would be interesting to investigate how the margin becomes different by $MR^2$ after training when visualized in a 2-dimensional space together with the features and classifier. (This is purely an optional comment stemming from personal curiosity and does not affect my decision.)
> >
> > **W3.**
> > Thank you for providing detailed comparative explanations of the proposed methods. I agree that the authors’ contributions are distinct from existing approaches discussed in the related works. However, even if prior methods do not explicitly separate easy and hard classes when learning margins, they may still exhibit some degree of effectiveness. In other words, previous approaches might have indirectly mitigated the class disparity issue that this paper aims to address, and thus there may be some theoretical or empirical overlap with $MR^2$. My intention in requesting such comparisons was to encourage a more generalized presentation of the theoretical contribution of $MR^2$, not to disagree with the contribution itself.
> >
> > **Q2 & Q3.**
> > Thank you for the additional information. Just to clarify—my question about how many times you conducted the experiments was purely out of curiosity and not due to any concerns about the authors’ code of ethics. I hope this clears up any possible misunderstanding.
> >
> > ### Closing Remarks
> >
> > ---
> >
> > Considering the revisions above, I will maintain my positive score, as the authors have adequately addressed the concerns through their updates. However, I have not yet thoroughly reviewed the comments from the other reviewers and the corresponding author responses, so I would like to finalize my decision toward the end of the discussion/rebuttal period. I will complete my review as soon as possible and notify you if my decision changes.

---

> > > ### Author Response · Authors · 2025-11-27
> > >
> > > Thank you for the thoughtful follow-up comments and for carefully reviewing our rebuttal. We are glad that our additional analyses and experiments addressed your concerns regarding W1–W3 and Q1–Q3.
> > >
> > > Regarding your interest in visualizing the margin geometry, Figure 7 provides the evolution of feature variance, showing that MR$^2$ steadily narrows the compactness gap between easy and hard classes. In the final version, we will further incorporate a 2-D t-SNE visualization of the features and classifier to complement this analysis.
> > >
> > > We thank you again for the positive evaluation and for recognizing the additional effort put into the revision, and we look forward to your final decision once the discussion period concludes.

---

> ### Author Response · Authors · 2025-12-02
> **Updated 2-D visualization**
>
> Dear Reviewer 5Lxp
>
> For your personal curiosity. In the revised version, we have added a t-SNE visualization (Fig. 11) of features for randomly selected hard CIFAR-100 classes, comparing standard cross-entropy with our MR$^2$ loss. The plot shows that, with MR$^2$, hard-class features become noticeably more compact and better separated across classes, which is consistent with our quantitative margin analysis and further supports our interpretation of how MR$^2$ reshapes the margin geometry.
>
> Best,
>
> Authors #6560

---

### Official Review · Reviewer_aWLX · 2025-11-02

**Soundness:** 2
**Presentation:** 3
**Contribution:** 2
**Rating:** 4
**Confidence:** 4

**Summary:**

This paper proposes a Margin loss for the balanced datasets, which tries to balance the variance in hard classes by using a larger margin. The margin loss is theoretically explained by proving a bound on the Rademacher complexity. Experiments have been provided on numerous datasets such as CIFAR-100, Pets, ImageNet etc.

**Strengths:**

1. Paper introduces a theoretically motivated solution to the problem.

2. The paper is well structured with relevant experiments.

**Weaknesses:**

1. Experiments are done on an older setup: I find that the experiments are done on older SOTA setups. The newer setups, like Sharpness Aware Minimization (SAM) [R1], WideResNets, have not been considered for comparison. Hence, the performance reported for datasets like CIFAR-10 and ImageNet is much lower than the current SoTA. Further, the margin-based algorithms like LDAM, compared with MR2, perform much better when compared to SAM [R2].

2. Missing Comparison: There are some contrastive learning methods that use Supervised Labels, like SupCon. As MR2 also uses lables it would be better to compare with them. It would be great to see some comparison with SotA (State-of-the-Art) frameworks in mind.

3. Novelty: I find the novelty of the paper to be a little limited, as I found an existing paper that talks about balancing the feature across classes [R3]. Further, a lot of the theoretical ideas are similar to those presented in Cortes et al. 2025 [R4].

[R1] Foret, Pierre, et al. "Sharpness-aware minimization for efficiently improving generalization." arXiv preprint arXiv:2010.01412 (2020).

[R2] Rangwani, Harsh, Sumukh K. Aithal, and Mayank Mishra. "Escaping saddle points for effective generalization on class-imbalanced data." Advances in Neural Information Processing Systems 35 (2022): 22791-22805.

[R3] Zhong, Ke, et al. "Class-Center-Based Self-Knowledge Distillation: A Simple Method to Reduce Intra-Class Variance." Applied Sciences 14.16 (2024): 7022.

[R4] Cortes, Corinna, et al. "Balancing the scales: A theoretical and algorithmic framework for learning from imbalanced data." arXiv preprint arXiv:2502.10381 (2025).

**Questions:**

Could the authors explain the difference in theoretical ideas from Cortes et al. (2025)?

---

> ### Author Response · Authors · 2025-11-21
> **Response to Reviewer aWLX**
>
> > W1. The newer setups, like Sharpness Aware Minimization (SAM) [R1], WideResNets, have not been considered for comparison.
>
> **Reply:** We thank the reviewer for the helpful suggestion. In the revised version, we have added experiments using SAM (Table 1) and WideResNets  (Table 2). These experiments show that MR$^2$ remains effective under stronger training pipelines: it works well with WideResNets and, importantly, substantially outperforms SAM in reducing the class-wise performance gap, even when SAM improves the overall accuracy.
>
> > W2. Missing Comparison: There are some contrastive learning methods that use Supervised Labels, like SupCon. As MR2 also uses lables it would be better to compare with them.
>
> **Reply:** We would like to clarify that *we did include Supervised Contrastive Learning (SupCon (Khosla et al., 2020)) in our comparisons*. In Table 1, we refer to it by another abbreviation SCL, which corresponds exactly to the SupCon method mentioned in the review. Our results therefore already include the requested comparison. In addition, we also include DRL, another supervised contrastive learning method, in our main experiments.
>
> We apologize that the abbreviation SCL may have caused confusion, and we have changed the naming to SupCon in the revised version.
>
> > W3.1. Comparison with Zhong et al.
>
> We thank the reviewer for pointing us to [R3]. While both [R3] and our work encourage compact features, **the goals and formulations are fundamentally different**:
>
> - **Problem focus.** [R3] aims to globally reduce intra-class variance in order to improve overall classification accuracy. It does not target class-wise performance disparity or the imbalance between “easy” and “hard” classes.  In contrast, our work is explicitly motivated by reducing class-wise accuracy gaps under class-balanced data (Fig. 1), and all our analyses and metrics are centered around this fairness-oriented objective.
> - **Method formulation.** [R3] proposes center self-distillation (CSD): per-class feature centers are updated online and their predictions are used as soft targets. This is an output-level self-distillation scheme that implicitly affects the feature distribution.
> Our method MR$^2$, instead, introduces two explicit margin-based regularizers (Eq. (1)–(2)). Thus, MR$^2$ is not a distillation method, but a margin-geometry–driven regularization that explicitly links per-class spread to decision margins.
> - **Balancing across classes vs. absolute shrinkage.** CSD in [R3] mainly aims to make all features of the same class aggregate around a center; it does not distinguish or balance the spreads of easy vs. hard classes. Our margin losses (logits and representation) are designed to reduce the imbalance of spreads across classes, so that hard classes with overly diverse features are regularized more strongly. This “balancing” is crucial for closing class-wise accuracy gaps, which is outside the scope of [R3].
> - **Theoretical perspective.** [R3] provides an empirical analysis of feature dynamics under self-distillation, and proposes the heuristic solution. However, it does not connect its loss to  generalization bounds. In contrast, our MR$^2$ is derived from a Rademacher complexity–based analysis (Proposition 1, Corollary 1), which shows how the adaptive logit margins and representation margins jointly tighten the generalization bound for classes with larger spread. This theoretical link to margin geometry is unique to our work.
>
> - **Small-scale vs. large-scale evaluation.** [R3] conducts experiments only on small-scale datasets using ConvNets, with at most 200 classes and about 100K training samples. By contrast, our method is validated on ImageNet with 1,000 classes and 1.2M training images, across both ConvNet and ViT architectures as well as different pretrained backbones.
>
> We have included the comparison in the revised version, Lines 906-936

---

> ### Author Response · Authors · 2025-11-21
>
> > W.1 & Q.1 Comparison with Cortes  et al.
>
> We thank the reviewer for pointing us to Cortes et al. (2025) [R4]. While both works use margin-based analyses, **the objectives, theoretical focus and implementation are substantially different**:
>
>
>
> - **Opposite objectives.** Cortes et al. aim to improve generalization under long-tailed label distributions: their class-dependent margins are tailored to perform well on the *given* imbalanced distribution, without attempting to equalize class-wise errors (majority classes can retain larger effective margins, sacrificing minority classes performance to achieve better overall performance, explained below). In contrast, our work is explicitly designed to *reduce class-wise performance disparity* by balancing margins across classes.
>
> - **Margin bounds are different.** Due to these opposite objectives, the resulting generalization bounds are inherently different.
> Cortes et al. derive bounds under imbalanced label distributions, where majority classes receive larger weights in the complexity term, so the induced optimal margins naturally favor majority classes. By contrast, our bound is for balanced performance: each class contributes symmetrically, and the derived margin regularizer explicitly enlarges margins for high-spread (typically hard) classes, thereby reducing class-wise performance disparity.
>
> - **Resulting margin implementations differ.** Because the margin bounds are different, the procedures for choosing class-wise margins also diverge. Cortes et al. obtain their “ideal’’ margins via a Rényi-divergence–based objective and express them in terms of a worst-case $\ell_2$ radius $r_k$ (Page 9), which can be unstable to estimate in practice. Consistent with this, Cortes et al. ultimately treat the class-wise margins as hyperparameters tuned on a validation set, whose search complexity scales with the number of classes (in their implementation, the theoretically derived formulas are not applied). This reliance on per-class margin search likely makes it challenging to scale their implementation to large benchmarks such as ImageNet-1K. In contrast, we derive our adaptive margins by directly minimizing the bound with a simple Lagrangian formulation, leading to closed-form updates in terms of *class-average* feature norms, which are more robust and readily applicable to large-scale settings.
>
> - **Uniqueness of our general norm bounds (Proposition 2).** Our core theoretical result extends the margin-based generalization bound to the case of *general norm* for $p\ge 1$ on the representations, which is not covered in Cortes et al. [R4].
> This is crucial for modern pretrained models where features are often $\ell_2$-normalized (e.g., CLIP, MoCo).
> In such settings, feature norms are effectively fixed (e.g., $||\mu_y||^2 + ||s_y||^2 = 1$ for all classes), so the class-wise generalization bound in~[R4] no longer provides informative class-dependent distinctions, as all classes become indistinguishable under their bound.
>
> We have included the comparison in the revised version, Lines 938-971.

---

### Official Review · Reviewer_MUKn · 2025-11-03

**Soundness:** 3
**Presentation:** 3
**Contribution:** 3
**Rating:** 4
**Confidence:** 5

**Summary:**

In this work, the authors propose a margin-based regularization framework to mitigate performance disparities between “hard” and “easy” classes that persist even when the training data are class-balanced. They begin by empirically showing that class-wise accuracy differences correlate with disparities in feature variance and margin distribution across classes. Building on this observation, the authors introduce a modification to the cross-entropy loss composed of two complementary terms: a logit-level margin regularizer that adaptively scales per-class margins based on feature variability, and a representation-level regularizer that enforces intra-class compactness by minimizing the spread of embeddings around each class mean.

The paper then provides theoretical analysis, first deriving an upper bound on the standard cross-entropy loss via their margin-regularized formulation, and subsequently identifying the optimal hyperparameter configuration that minimizes this bound. The analysis is further extended to normalized embedding spaces—commonly used in contrastive and transformer architectures—showing similar generalization guarantees and optimal margin design.

Finally, the authors conduct extensive experiments, training from scratch on smaller datasets such as CIFAR-100 and fine-tuning or linearly probing large pre-trained models on ImageNet and other fine-grained datasets. Across all cases, the proposed method improves the accuracy of “hard” classes without degrading, and sometimes even enhancing, performance on “medium” and “easy” classes. The authors also include ablations showing the robustness of their results to the choice of hyperparameters.

**Strengths:**

The paper is clearly written, and the ideas it explores are relevant to our broader understanding of optimization. The theoretical outline is well presented and easy to follow, and the experimental setup is generally sound and consistent with prior work. The proposed approach makes a meaningful contribution by improving performance on hard classes without hurting the easier ones, leading to a more balanced overall accuracy across classes.

**Weaknesses:**

**W1)**: I believe this work, given its focus on margin geometry and embedding compactness, overlooks a closely related and highly relevant area known as Neural Collapse (Papyan et al., 2020). This phenomenon shows that in over-parameterized networks—such as those considered in this paper—class embeddings tend to collapse to a single prototype per class with maximal inter-class separation as training progresses. Subsequent works have analyzed Neural Collapse under class imbalance (Behnia et al., 2023), supervised contrastive losses (Kini et al., 2023), and in relation to generalization and transfer learning (Galanti et al., 2021). Even more recently, Neural Collapse principles have been extended to other domains such as federated learning, where margin adjustment inspired by NC has been proposed to address class imbalance (Li et al., 2025).

From a theoretical standpoint, if we follow the Neural Collapse arguments, the regime where $ ||s_y|| \to 0$  would eventually reduce the proposed loss to standard cross-entropy, since the representation compactness term becomes inactive and the per-class margins converge. This connection appears important but is not acknowledged or discussed in the paper. While the authors do a solid job referencing prior work on logit adjustment and class-wise accuracy imbalance, they omit this major line of research that directly studies how margin-controlled and embedding-controlled objectives influence model geometry and generalization. I would be interested to see the authors comment on this relation during the discussion period, given the conceptual overlap between their margin regularization intuition and the Neural Collapse framework. Much of the NC work struggles with how to relate the geometry of features during training to generalization and much of the theoretical upper bound for optimal geometry struggle in practice on improving test performance.

**W2)**: While the authors primarily focus on the margins of hard versus easy classes, their evaluation relies solely on accuracy as the performance metric. Although the reported improvements are clear and likely related to the margin and embedding-spread–based logit adjustments, additional experimental analysis would help solidify these claims. In particular, it would be useful to visualize how the margins and the $s_y$ parameter evolve during training under their objective. Without such analysis, it remains unclear how much of the observed improvements stem from the proposed margin regularization itself versus other implicit effects of regularization.

**Questions:**

**Q1:**
In line 289, the authors mention that overtly increasing the value of the margin parameter is not desirable and may lead to a higher empirical margin. Could the authors expand on this point and provide a more precise theoretical explanation? I find this statement somewhat unclear. A related follow-up question is how, according to the proposed loss terms, simply reducing the inter-class spread parameter $s_y$ to zero would not already minimize the overall loss objective.

**Q2:**
Why do the authors not evaluate their method under class imbalance settings? From my reading of the results, the imbalance scenario should conceptually produce a similar difference in margin distribution as the one discussed here. This relationship has been explicitly examined in the Neural Collapse literature (Fang et al., 2021). Furthermore, given that the ImageNet dataset already exhibits some class imbalance, wouldn’t the improvements observed on it indirectly reflect performance under imbalance? Am I correct to assume that classical ImageNet is indeed imbalanced?

**Q3:**
Were the experiments involving other architectures and loss formulations conducted only in the fine-tuning stage, rather than full training? If so, do the authors believe that training from scratch on harder benchmarks such as ImageNet with their loss formulation would yield noticeable improvements?

**Q4:**
Regarding the representation margin term, is this component designed to enforce a balance on the value of $\bar{s}$—that is, to keep it small but non-zero? Could the authors elaborate on this point or clarify whether any part of their theoretical analysis addresses the optimal regime or equilibrium value of this term?

---

> ### Author Response · Authors · 2025-11-21
> **Response to Reviewer MUKn**
>
> > W1. ... I would be interested to see the authors comment on this relation during the discussion period, given the conceptual overlap between their margin regularization intuition and the Neural Collapse framework...
>
> **Reply**: We thank the reviewer for pointing this out. Neural Collapse (NC) is relevant to our problem, and discussing this link provides valuable additional insight. Let us first revisit the empirical property of Neural Collapse (NC) that is most relevant to our problem. During the terminal phase of training (TPT) of deep networks:
>
> - **Variability collapse of last-layer features.** Within-class feature variance collapses and features concentrate tightly around their respective class means.
>
> In our experiments, we observe that the idealized pattern do not fully emerge on large-scale datasets, such as ImageNet.
> Our observations are consistent with the empirical study of Cui et al. [R1], who measures feature variability. The results show that the within-class feature variations remain non-negligible, (and we also empirically verify that the average ratio between variance norm and mean feature $ \mathrm{Avg}_y[||\hat{\mathbf{s}}_y|| / ||\hat{\boldsymbol{\mu}}_y||] \approx 0.3285$), and these variations are imbalanced between easy and hard classes (Fig.8 in our revised version and Sec. 4.3 and Fig. 2 in Cui et al. [R1]).
>
> The gap between the ideal NC behavior and the empirical geometry observed on datasets such as ImageNet can be attributed to at least two factors:
>
> First, **task hardness**. The difficulty of a classification problem depends on the number of classes, the sample size, and the intrinsic complexity of the visual concepts. Inspecting Fig. 6 (first row) in Papyan et al. [R2], one observes that as task difficulty increases, the within-class feature variation ceases to collapse toward zero: the magnitude of the within-class to between-class variance is on the order of $10^{-4}$ for the easiest task (MNIST), but exceeds $1$ for the most challenging subsampled ImageNet setting. To further illustrate that feature variation collapse tends to appear only on simpler datasets, we repeat the same analysis on MNIST, using the same training configuration as in ImageNet (Fig.8 in our revised version) but with a ResNet-18 backbone. The results appear in Table below:
>
> **Table 1.** Per-class $||\hat{\mathbf{s}}_y||_2$ (var.) and $ ||\hat{\boldsymbol{\mu}}_y||_2$ (mean) on MNIST.
>
> | class id              | 1      | 2      | 3      | 4      | 5      | 6      | 7      | 8      | 9      | 10     |
> |------------|-------|-------|-------|-------|-------|-------|-------|-------|-------|-------|
> |  var | 0.0041 | 0.0032 | 0.0048 | 0.0051 | 0.0056 | 0.0063 | 0.0040 | 0.0056 | 0.0056 | 0.0063 |
> |  mean | 6.6982 | 6.7474 | 6.5227 | 6.8819 | 6.7275 | 6.9866 | 6.4091 | 6.4091 | 6.7820 | 6.6932 |
>
> On MNIST, the variability collapse of features  does emerge: the ratio
> $ \mathrm{Avg}_y[||\hat{\mathbf{s}}_y|| / ||\hat{\boldsymbol{\mu}}_y||] $ is approximately $7.5 \times 10^{-4}$.
> However, recall that on ImageNet this ratio is much larger,
> $ \mathrm{Avg}_y[||\hat{\mathbf{s}}_y|| / ||\hat{\boldsymbol{\mu}}_y||]  \approx 0.3285$, indicating that the collapse of features does not hold in this harder, large-scale setting.
> Moreover, Papyan et al. [R2] use only 600 training samples per class—roughly half of the full ImageNet training size—yet still observe substantial residual within-class variability.
> This suggests that non-vanishing $||\hat{\mathbf{s}}_y|| $ is expected for large-scale, high-complexity datasets.
>
> Second, **practical regularization**. Modern training pipelines routinely employ regularization strategies that were not considered in Papyan et al. [R2] and that hinder complete variability collapse. *Early stopping* prevents networks from entering the extremely long TPT required for NC to fully emerge, since training is typically stopped according to validation performance rather than continued for many additional epochs. In addition, *strong data augmentation* substantially increases the effective difficulty of the task by injecting extra intra-class variation, while Papyan et al. [R2] train without augmentation.
>
> Taken together, these observations suggest that NC should not be viewed as a **golden rule** for deep model deployment; its manifestation depends strongly on task hardness and regularization.
> These factors naturally inhibit complete within-class variability collapse in contemporary large-scale settings, and further justify modeling and regularizing the residual, class-dependent spreads $||\hat{\mathbf{s}}_y||$ as we do in this work.
>
> We further provided a detailed discussion in Section A, Lines 782-837.
>
> [R1] Cui et al. Classes are not equal: An empirical study on image recognition fairness. In CVPR, 2024.
>
> [R2] Papyan et al. Prevalence of neural collapse during the terminal phase of deep learning training. PNAS, 2020.

---

> ### Author Response · Authors · 2025-11-21
>
> > W2 In particular, it would be useful to visualize how the margins and the $s_y$ parameter evolve during training under their objective.
>
> **Reply**. We thank the reviewer for the helpful suggestion. In response, we record the evolution of $||\mathbf{s}_y||$ during training on ImageNet, with and without our loss. The corresponding curves have been added to the revised manuscript as Fig. 7.
> The contrast is clear: our MR$^2$ consistently compresses the gap between easy and hard classes throughout training, whereas the baseline exhibits a steadily widening disparity. This demonstrates that MR$^2$ effectively suppresses the growth of class-dependent feature spreads.
>
> | Method | Output-level margin Avg. ↑ | Output-level margin Easy ↑ | Output-level margin Med. ↑ | Output-level margin Hard ↑ | Classifier-level margin ↑ |
> |--------|----------------|---------------|------|----|------------|
> | ERM    | 0.158          | 0.236         | 0.144         | 0.093          | 58.4       |
> | MR$^2$    | 0.373          | 0.435         | 0.396         | 0.289          | 81.8       |
>
> In Table 4 of the revised version (or the table above), we also study the effect of our MR$^2$ on output-level and classifier-level margins (formal definition of the two margins is in the main paper). As in Table 4, MR$^2$ significantly enlarges output-level margins across all subsets, reduces the margin gap between easy and hard class, and raises the classifier-level margin from 58.4 to 81.8.
>
> > Q1.1. Explanation for ``overly increasing the value of the margin parameter is not desirable.''
>
> **Reply**: Recall that the empirical margin risk is defined as empirical mean of $\ell_{\gamma}(f,\mathbf{x},y)$, and $\ell_{\gamma}(f,\mathbf{x},y)$ is increasing with respect to $\gamma$ (Definition 1).
> As illustrated in Figure 2, when the margin parameter becomes excessively large ($\gamma \rightarrow \infty$),
> the loss $\ell_{\gamma}(f,\mathbf{x},y)$ approaches $1$ for almost all samples—i.e., every sample incurs the worst loss—leading to an inflated empirical risk (the same behavior for its cross-entropy surrogate). We have incorporated this explanation into the revised version Lines 307–309.
>
> > Q1.2 How simply reducing the inter-class spread parameter $s_y$ to zero.
>
> **Reply**: As discussed in our response to W1, within-class feature variations are non-negligible in practice, and whether they can be reduced toward 0 depends heavily on task difficulty and common training regularization techniques. Moreover, these variations are inherently imbalanced across easy and hard classes: while some easy classes may be pushed closer to zero variability, it is substantially harder for hard classes to do so. In contrast, our proposed objective is minimized more effectively by promoting a **more balanced spreads** rather than a uniform collapse to zero. We have incorporated this discussion in the revised version Lines 839-845.
>
> > Q2.1 Why do the authors not evaluate their method under class imbalance settings?
>
> **Reply**: In the main experiments, we omit evaluation under class-imbalance settings, because our goal is to disentangle two fundamentally different sources of class-wise performance disparity:
> - imbalanced class numbers
> - imbalanced class feature spreads.
>
> Valuating on imbalanced datasets would entangle these effects and obscure the specific phenomenon we aim to study.
>
> For class-imbalance scenarios, extensive prior work already provides well-established solutions. These methods typically attribute class-wise disparity to shifted decision boundaries and thus introduce classifier-level corrections (often termed decoupled learning in long-tail recognition), such as freezing the backbone and retraining or reweighting classifier weights (Kang et al. [R3]), or adjusting classifier biases/margins (Menon et al. [R4]). Notably, Menon et al. [R4] show that a Fisher-consistent solution for balanced error is achieved by subtracting the log class prior from the predicted logits (the logit-adjustment method, LA).
>
> **Table 2. Applying our  MR$^2$ to CIFAR-100-IB-100. (class-imbalanced dataset)**
> | Model | Overall | Easy  | Medium | Hard  |
> |--- |---|---|--- |---|
> | ERM | 37.7    | 69.6  | 38.9   | 5.4 |
> | LA| 41.9    | 66.9  | 43.9   | 15.6  |
> | LA + MR$^2$ | 44.2    | 68.1  | 46.3   | 18.9  |
>
> These classifier-level techniques are orthogonal to our approach and can be combined with MR$^2$  in a modular way. To demonstrate this, we incorporate LA into MR$^2$ and evaluate the hybrid method on the standard long-tail benchmark CIFAR-100-IB-100 using ResNet-32. The results in Table above show that incorporating  MR$^2$ with the imbalanced-learning method LA further improves overall accuracy and reduces performance disparity. We have added these experiments in Lines 854-859 in the revised version.
>
> [R3] Kang et al. Decoupling representation and classifier for long-tailed recognition. In ICLR, 2020.
>
> [R4] Menon et al. Long-tail learning via logit adjustment. In ICLR, 2021

---

> ### Author Response · Authors · 2025-11-21
>
> > Q2.2  Am I correct to assume that classical ImageNet is indeed imbalanced?
>
> **Reply**: We use the ImageNet-2012 training set, whose class distribution is only *mildly* imbalanced: about 678 classes contain around 1200 samples, 97 classes have 1000 samples, 157 classes have 800 samples, and 68 classes contain 700--799 samples. The resulting imbalance ratio (defined as the ratio between the largest and smallest class sizes; higher values indicate more severe imbalance) is below 2, which is negligible compared to standard imbalanced benchmarks such as ImageNet-LT (an imbalanced version of ImageNet), where the imbalance ratio reaches 256 or iNaturalist-2018 where the imbalance ratio is 512. Under such a mild imbalance, class-wise performance disparity is not primarily driven by sample-count imbalance.
>
> > Q3 Were the experiments involving other architectures and loss formulations conducted only in the fine-tuning stage, rather than full training? If so, do the authors believe that training from scratch on harder benchmarks such as ImageNet with their loss formulation would yield noticeable improvements?
>
> **Reply**: All experiments on CIFAR-100—including the results in Table 1, Table 2, and the ablation studies—were trained from scratch, rather than fine-tuned. This ensures that the comparisons across architectures and loss formulations are fair and not influenced by pretrained initialization.
>
> Our loss formulation also functions well with train from scratch on harder ImageNet. We adopt the pre-train–then–fine-tune paradigm because it is the prevailing practice in modern large-scale vision training, and we aim for our method to be compatible with current practical workflows rather than tied exclusively to training from scratch. To further address the reviewer’s concern, we additionally train ResNet-50 *from scratch* on ImageNet for 300 epochs using our loss. The results are shown in the following table.
> | Model        | Overall | Easy                | Medium              | Hard                |
> |-------------|---------|---------------------|---------------------|---------------------|
> | RN-50 train from scratch   | 71.7    | 88.5                | 74.1                | 52.6                |
> | + MR$^2$    | 74.2    | 89.9 **(+1.4)**     | 76.9 **(+2.8)**     | 55.9 **(+3.3)**     |
>
> These results confirm that our method remains effective even when training from scratch on ImageNet. We have also included the experiments in the revised paper (Table 2).
>
> > Q4 Regarding the representation margin term, is this component designed to enforce a balance on the value of $\bar{s}$—that is, to keep it small but non-zero? Could the authors elaborate on this point or clarify whether any part of their theoretical analysis addresses the optimal regime or equilibrium value of this term?
>
> **Reply:** Yes, our component encourages the model to reach a *balanced and data-dependent equilibrium* rather than forcing $s_y$ to be zero. Our theoretical analysis does not specify an analytically value of $s_y$; as shown in Eq.~(3), once all classes attain the same spread, the corresponding term in the objective converges to zero. The eventual equilibrium value of $s_y$ depends on the intrinsic hardness of the task, as discussed in our response to W1.
>
> For example, Papyan et al.[R2] demonstrate that for easier datasets such as MNIST or FashionMNIST, the NC property can fully emerge and $s_y$ can approach zero. In contrast, for harder benchmarks such as ImageNet, NC does not hold empirically, and the equilibrium value of $s_y$ is non-zero and cannot be determined analytically. Our formulation is therefore designed to balance the class-wise spreads rather than collapse them to zero.

---

### Author Response · Authors · 2025-11-21
**Response to All Reviewers**

Dear Program Chair, Senior Area Chair, Area Chair, and Reviewers,

We thank the reviewers for their thoughtful feedback. We are encourage that they found our paper clearly written (MUKn, aWLX, 5Lxp), theoretically well grounded (MUKn, aWLX, 5Lxp, dj7P)), and supported by comprehensive experiments (MUKn, aWLX, 5Lxp, dj7P). We appreciate their recognition that our method advances optimization understanding, improves hard-class performance without affecting easy ones (MUKn), and tackles the overlooked inter-class disparity problem (5Lxp, dj7P).

In addition, we have uploaded a new revision of our paper (modifications are in blue text), in which we incorporate most of the comments from the reviewers. Specifically, we:

- Added a discussion of hyperspherical learning methods in Section 2, Lines 100–102 (5Lxp).
- Added a discussion of Neural Collapse in Section 3.1, Lines 207-214 and Section A, Lines 782-837 (MUKn).
- Added a discussion of temperature scaling in Section 3.1, Lines 214-231 (dj7P).
- Added an explanation of why overly increasing $\gamma$ is undesirable in Section 4.1, Lines 307–309 (MUKn).
- Added more baselines (LGM, CMIC-DL, SAM, DFL) in Section 5.1, Table 1 (aWLX, dj7P).
- Added results of WideResNet (aWLX)  and ImageNet with training from scratch (dj7P, MUKn) in Table 2.
- Added a visualization of the evolution of $||\hat{\mathbf{s}}_y||_2$ in Figure 7 (MUKn).
- Added a study on the effect of MR$^2$ on output-level and classifier-level margins in Table 4 (MUKn, 5Lxp).
- Added a discussion of whether making $\gamma$ trainable would be beneficial in Section A, Lines 758-762 (dj7P).
- Added experiments with advanced data augmentations in Section A, Lines 775-781 (dj7P).
- Added an explanation of why simply reducing  $||\hat{\mathbf{s}}_y||_2$ is less effective in Section A, Lines 839-845 (MUKn).
- Added an explanation on the limited impact of variations in   $||\hat{\boldsymbol{\mu}}_y||_2$ in Section A, Lines 846-853 (dj7P).
- Added an experiment of applying our MR$^2$ to class-imbalanced settings in Section A, Lines 855-895 (MUKn).
- Added a discussion of the limitation and future direction of our work in Section A, Lines 896-905 (5Lxp).
- Added a discussion of Zhong et al. 2024 and Cortes et al. 2025 in Section A, Lines 905-971 (aWLX).
- Added ablation studies of $\bar{c}$ and $\lambda$ on ImageNet in Figure 9-10 (5Lxp).
- Added standard deviations  across diverse model architectures on CIFAR-10 and ImageNet in Table 15 (5Lxp).

Best regards,

Authors of Paper #6560

---

### Comment · Area_Chair_KWxo · 2025-11-27

Dear Reviewers,

Thank you for the time and effort you have dedicated to reviewing this paper and providing thoughtful feedback. The authors have now submitted their responses to your comments. I kindly ask that you engage in the discussion with them and assess whether your concerns and questions have been fully addressed before the December 2 deadline.

Please also keep in mind that the author–reviewer relationship is reciprocal; the engagement you offer here reflects the same level of consideration you would expect when you are on the author side.

Thank you for your continued support and cooperation.

Best regards,
AC

---

### Author Response · Authors · 2025-12-03
**Rebuttal Summary for Newly Assigned AC**

Dear re-assigned AC,

Thank you for your efforts in coordinating the review process. Below is a concise summary of the reviews and our rebuttal.

### Before rebuttal
---
All reviewers (`MUKn, aWLX, 5Lxp, dj7P`) find our paper **theoretically well grounded** with **comprehensive experiments**. `MUKn` highlights that our method **advances optimization understanding**, improves hard-class performance without affecting easy ones. `5Lxp` and `dj7P`   recognize that we tackle the **overlooked inter-class disparity problem.**

### During rebuttal
---
`dj7P` and `5Lxp`—all concerns addressed; both maintained positive scores.

`MUKn` and `aWLX` did not respond before the Openreview issue. Our key responses are summarized below.

Reviewer `MUKn`

* **W1 (Relation to NC)**:  We show NC barely holds in our setting and provide empirical evidence and explanations.

* **W2 (Margin visualization)**: Added in revised Figure 7 and Table 4.

* **Q1: (Why overly large margin parameter is not desirable)**:  Our objective is monotonically increasing with the parameter, explained theoretically.

* **Q2 (Imbalanced setting)**: Results added in Table 7.

* **Q3 (Training from scratch on ImageNet)**: Added in Table 2(b).

* **Q4: (Equilibrium value of $\bar{s}$)** The value is data-dependent based on the hardness of the task.

Reviewer `aWLX`

* **W1: (Experiments of SAM and WideResNet)**.  Added in revised Table 1 and Table 2.

* **W2: (Experiments of SupCon)** We did include SupCon in our original manuscript.

* **W3 & Q1 (Comparison with Zhong et al. & Cortes et al.)**: We explain that our goals, formulations, and theoretical focus differ fundamentally from both works.

---
Thank you again for your time and efforts.

Best regards,

Authors #6560

---

### Meta-Review · Area_Chair_8CdY · 2026-01-04

**Summary:**

The paper is clearly written, theoretically motivated, and addresses the important problem of class imbalance by improving performance on hard classes without harming easier ones, with solid theoretical analysis and extensive experiments supporting its claims. However, reviewers raised concerns that the work overlooks the Neural Collapse phenomenon and its relation to margin regularization, relies primarily on accuracy without analyzing margin evolution, introduces additional hyperparameters requiring careful tuning, lacks comparisons to newer state-of-the-art methods and augmentation strategies, shows some overlap with prior work in margin and embedding control, and could benefit from training-from-scratch experiments and further verification of class margin behavior across datasets.

**Reviewer Concerns:**

The authors addressed all key issues raised by the reviewers, including the lack of detailed theoretical analysis, the relationship between the proposed method and neural collapse phenomena, the function and impact of the boundary regularization mechanism, the role and effectiveness of each component, the insights provided by ablation studies, the selection and tuning of hyperparameters, the scalability and generalization capabilities of the experiments across different datasets and architectures, and comparisons with existing state-of-the-art methods. The authors provided detailed explanations, clarified the theoretical motivations, and validated their claims through extensive supplementary experiments, visualizations, and analyses. Therefore, all major concerns raised during the review process have been fully and satisfactorily addressed.

**Reviewer Scores:**

Reviewers MUKn and aWLX are likely to increase their scores after seeing the authors’ thorough and well-prepared responses. Overall, the paper does not have any major issues, and the authors have addressed all reviewer concerns comprehensively and convincingly.

---

### Decision · Program_Chairs · 2026-01-26

Accept (Poster)